# Is Spurious Correlation Removal Always Learnable?

**Yibo Zhou**[1]  **Bo Li**[1 2]  **Hai-Miao Hu**[1 2 3]  **Hanzi Wang**[4]  **Xiaokang Zhang**[1]  **Ruifan Zhang**[1]

## Abstract

Invariant learning can fail even when the invariant structure is statistically identifiable. We show a conditional computational barrier: under a black-box samplable supervised sparse recovery primitive motivated by average-case sparse-recovery reductions, there exist *samplable* multi-environment instances with a one-dimensional predictive invariant subspace ($k = 1$) that are learnable with polynomial samples by exhaustive search, while any polynomial-time constant-accuracy recovery algorithm would contradict the primitive. We further quantify environment diversity by a separation parameter $\gamma$, which controls identifiability and the curvature of invariance objectives. Under sufficient diversity and local Gaussian regularity, the minimax risk is $\mathbb{E}[\mathrm{dist}(\hat{V}, V_{\mathrm{inv}})^2] = \Theta(k(d-k)/(n|\mathcal{E}|))$, and under label-induced shifts a phase transition occurs at $n^* \propto k(d-k)/(|\mathcal{E}|\gamma^2)$ with refined estimation error scaling proportional to $1/\gamma^2$. Synthetic and real datasets illustrate the predicted gaps and transitions and motivate simple diversity diagnostics.

## 1. Introduction

Machine learning models often exploit spurious correlations that hold in training data but fail under distribution shift (Sagawa et al., 2020). This issue appears in several related forms, where models must detect inputs outside the training distribution (Zhou, 2022; Zhang et al., 2024; Zhang & Hu, 2025a; Zhang et al., 2023a), and co-occurrence bias in image classification, where models may rely on unstable cor-

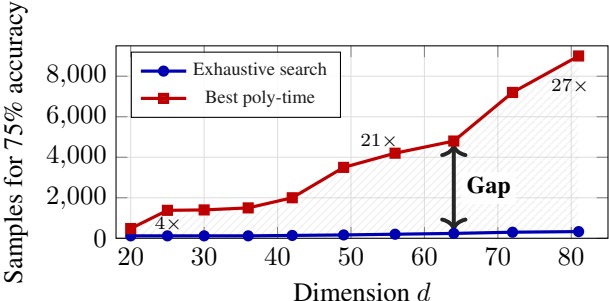

*Figure 1.* Compute–sample gap on synthetic instances: exhaustive search needs far fewer samples than the best polynomial-time method to reach the same feature-selection accuracy.

relations among classes rather than class-specific evidence (Zhou et al., 2023; Zhang & Hu, 2025b; Zhou et al., 2025a; 2024; 2025b; Zhang et al., 2023b). Invariant learning uses multi-environment data to target features whose predictive relation with the label is stable across environments (Arjovsky et al., 2019; Krueger et al., 2021). Theory shows that such invariant structure can be statistically identifiable under diversity assumptions (Rosenfeld et al., 2020), yet empirically, invariant methods sometimes help and sometimes do not (Gulrajani & Lopez-Paz, 2020; Koh et al., 2021). We ask whether this inconsistency can arise even in idealized settings where invariances are identifiable.

We answer yes by establishing a computational–statistical separation for spurious correlation removal. Under a black-box samplable supervised sparse recovery primitive, motivated by average-case sparse-recovery reductions but not claimed to follow directly from existing Planted-Clique-to-sparse-CCA reductions, we construct samplable multi-environment instances where the predictive invariant direction is identifiable as the unique maximizer of an invariance–predictivity population score and is recoverable with polynomially many samples by exhaustive (exponential-time) search, but any polynomial-time constant-accuracy recovery algorithm at comparable sample sizes would contradict the primitive (Theorem 3.8). The construction maps the black-box sparse primitive to labeled multi-environment samples without accessing the hidden support.

A second message is that the number of environments matters less than their *diversity*. We quantify diversity by a sepa-

---

[1]Beijing Key Laboratory of Digital Media, School of Computer Science and Engineering, Beihang University, Beijing 100191, China [2]State Key Laboratory of Virtual Reality Technology and Systems, Beihang University, Beijing 100191, China [3]Hangzhou Innovation Institute of Beihang University, Hangzhou 310051, China [4]Fujian Key Laboratory of Sensing and Computing for Smart City, Xiamen University, Xiamen 361005, China. Correspondence to: Hai-Miao Hu <hu@buaa.edu.cn>.

*Proceedings of the 43rd International Conference on Machine Learning*, Seoul, South Korea. PMLR 306, 2026. Copyright 2026 by the author(s).

ration parameter $\gamma$ measuring how differently environments perturb spurious correlations (Definition 2.3). When $\gamma = 0$ (i.e., environments do not induce distinguishable variation in spurious conditional means), the invariant subspace can be unidentifiable even with unlimited data (Corollary 4.7). When $\gamma > 0$ and the environment-induced spurious shifts are sufficiently *directionally rich* (Proposition 4.4), recovery becomes possible and the required sample size decreases as diversity increases; in a structured label-induced shift regime the critical scaling is proportional to $1/\gamma^2$ (Theorem 4.15). This predicts that a few diverse environments can be more informative than many similar ones.

We make four contributions. First, under a black-box samplable supervised sparse-recovery primitive, we construct samplable multi-environment instances exhibiting a computational–statistical gap for invariant subspace recovery. Second, we quantify environment diversity through a separation parameter $\gamma$ and relate it to identifiability, minimax risk, and sample-complexity transitions. Third, we identify structural conditions under which polynomial-time methods achieve near-optimal rates. Finally, experiments on synthetic and standard spurious-correlation benchmarks illustrate compute–sample tradeoffs, diversity dependence, and practical diagnostics. Our hardness results are worst-case and conditional; the framework suggests estimating environment diversity before attributing invariant-learning failures to optimization or algorithmic limitations.

**Conflict of Interest Disclosure.** The authors declare that they have no financial conflicts of interest related to this work. In particular, this paper does not evaluate any model, dataset, product, or service developed by an organization that financially supports or employs any of the authors in a way that would constitute a conflict under the ICML peer-review ethics guidelines.

## 2. Problem Formulation

We model spurious correlation removal as invariant subspace recovery from multi-environment data, focusing on a linear-Gaussian setting. For a subspace $V \subseteq \mathbb{R}^d$, let $P_V$ be the orthogonal projector and $\mathrm{Gr}(k, d)$ the Grassmannian of $k$-dimensional subspaces.

**Definition 2.1** (SC Problem Instance). A *spurious correlation (SC) instance* consists of orthogonal subspaces $V_{\mathrm{inv}}, V_{\mathrm{sp}} \subseteq \mathbb{R}^d$ with $V_{\mathrm{inv}} \oplus V_{\mathrm{sp}} = \mathbb{R}^d$ and $\dim(V_{\mathrm{inv}}) = k$, and environments $\mathcal{E} = \{1, \ldots, |\mathcal{E}|\}$ with distributions $\{\mathbb{P}_e\}_{e \in \mathcal{E}}$ over $(X, Y) \in \mathbb{R}^d \times \mathbb{R}$ such that

$$X = X_{\mathrm{inv}} + X_{\mathrm{sp}}, \qquad X_{\mathrm{inv}} = P_{V_{\mathrm{inv}}} X, \qquad X_{\mathrm{sp}} = P_{V_{\mathrm{sp}}} X.$$

The instance satisfies that **(C1)** the conditional mechanism $\mathbb{P}_e(Y \mid X_{\mathrm{inv}})$ is identical across $e$, and **(C2)** $\mathbb{P}_e(X_{\mathrm{sp}} \mid X_{\mathrm{inv}})$ differs for some $e \neq e'$.

**Assumption 2.2** (Gaussian Model). For each $e \in \mathcal{E}$, $(X^{(e)}, Y^{(e)})$ is jointly Gaussian, $X^{(e)} \sim \mathcal{N}(\mu^{(e)}, \Sigma^{(e)})$, and the distribution of $P_{V_{\mathrm{inv}}} X^{(e)}$ is identical across environments. Moreover, the conditional mechanism $\mathbb{P}_e(Y \mid P_{V_{\mathrm{inv}}} X)$ is environment-invariant.

Under Assumption 2.2, the environment-invariance of $\mathbb{P}_e(Y \mid X_{\mathrm{inv}})$ means that the conditional law of $Y$ given $X_{\mathrm{inv}}$ is identical across environments. In the statistical analysis (Section 4) we add the standard linear-Gaussian structural assumption that $Y = \langle w^*, X_{\mathrm{inv}} \rangle + \epsilon$ with $w^* \in V_{\mathrm{inv}}$.

We quantify cross-environment variation in spurious features by a separation parameter.

**Definition 2.3** (Environment Separation). For $x \in \mathbb{R}^k$ and $j \in \{1, \ldots, d - k\}$, define $\mu_{e,j}(x) := \mathbb{E}_e[X_{\mathrm{sp},j} \mid X_{\mathrm{inv}} = x]$ and $\Delta_j(x) := \max_{e \in \mathcal{E}} \mu_{e,j}(x) - \min_{e \in \mathcal{E}} \mu_{e,j}(x)$. Let $M(x) := \max_{1 \leq j \leq d-k} \Delta_j(x)^2$. Define $\gamma := \left(\mathbb{E}[M(X_{\mathrm{inv}})]\right)^{1/2}$, where the expectation is under the common marginal of $X_{\mathrm{inv}}$ when it is environment-invariant, and otherwise under the averaged environment marginal. In the mean-shift setting where $\mathbb{E}_e[X_{\mathrm{sp}} \mid X_{\mathrm{inv}}] = \mu_{\mathrm{sp}}^{(e)}$, we have $\gamma = \max_{e,e' \in \mathcal{E}} \left\| \mu_{\mathrm{sp}}^{(e)} - \mu_{\mathrm{sp}}^{(e')} \right\|_\infty$.

*Remark* 2.4 (Coordinate separation versus directional diversity). The scalar $\gamma$ is a coordinate-level separation parameter. It is useful for axis-aligned features and for diagnostics in learned representations, but by itself it does not guarantee that every spurious direction is exposed by environment variation. Full subspace identifiability generally requires a directional richness condition. In the label-induced shift model, this role is played by the environment-difference matrix

$$\mathcal{F} = \sum_{e < e'} (\mu_{\mathrm{sp}}^{(e)} - \mu_{\mathrm{sp}}^{(e')})(\mu_{\mathrm{sp}}^{(e)} - \mu_{\mathrm{sp}}^{(e')})^\top,$$

and especially by $\lambda_{\min}(\mathcal{F})$. The phase-transition result therefore assumes $\lambda_{\min}(\mathcal{F}) \gtrsim \gamma^2$. Without such a directional condition, a large coordinate-wise separation may still leave some spurious directions nearly unperturbed.

**Definition 2.5** (Empirical Diversity Proxy). The separation $\gamma$ is coordinate-dependent and is intended for axis-aligned features or learned representations; rotated settings use spectral quantities such as $\lambda_{\min}(\mathcal{F})$. Empirically, we use

$$\hat{\gamma} := \mathrm{median}_j \left( \max_e \hat{\rho}_j^{(e)} - \min_e \hat{\rho}_j^{(e)} \right),$$

$$\hat{\rho}_j^{(e)} := \mathrm{Corr}(X_j^{(e)}, Y^{(e)}),$$

and write $\hat{\gamma}_{\mathrm{repr}}$ when computed on learned features.

Let $n_e$ be the sample size in environment $e$, $N := \sum_{e \in \mathcal{E}} n_e$, and $n_{\min} := \min_e n_e$. In the balanced case, $n_e = n$ and $N = n|\mathcal{E}|$.

**Goal and metrics.** Given samples from all environments, recover $V_{\text{inv}}$. We evaluate estimators $\hat{V}$ using the subspace distance $\text{dist}(V, V') := \|P_V - P_{V'}\|_F$; for axis-aligned subspaces we also report feature selection accuracy (fraction of true invariant coordinates among the selected top-$k$).

*Remark* 2.6 (Environment-wise sampling). We adopt the standard multi-environment setting where samples are independent across environments and within each environment are i.i.d. In particular, the hard family in Section 3 is constructed by running the primitive sampler $\mathcal{R}(A)$ independently per environment with fresh randomness.

## 3. The Computational-Statistical Gap

We show that spurious correlation removal can be statistically feasible yet computationally intractable in a conditional black-box sense. The computational lower bound in this section is for a samplable multi-environment wrapper around a supervised sparse recovery primitive, while the linear-Gaussian statistical theory is developed separately in Section 4.

*Hypothesis* 3.1 (Planted Clique Hardness). Let $G(m, 1/2)$ denote the Erdős–Rényi random graph on $m$ vertices with edge probability $1/2$. In the planted clique problem, we observe $G \sim G(m, 1/2)$ with a planted $\kappa$-clique on a uniformly random subset $S \subseteq [m]$, and the goal is to recover $S$. Planted Clique Hardness states that for $\kappa = m^{1/2-\delta}$ with any constant $\delta > 0$, no polynomial-time algorithm recovers $S$ with probability $1 - o(1)$ (Jerrum, 1992; Alon et al., 1998; Feige & Krauthgamer, 2000; Berthet & Rigollet, 2013).

Hypothesis 3.1 is included as motivation for average-case sparse-recovery hardness. The formal lower bound below is conditional on the black-box supervised sparse recovery primitive in Hypothesis 3.2; the key requirement for our reduction is samplability, namely that samples can be generated in randomized polynomial time from the primitive input without using the hidden sparse direction or its support.

### 3.1. A Black-Box Samplable Supervised Sparse Primitive

We use a standard average-case hardness template: planted-clique-based reductions motivate sparse predictive recovery tasks, such as sparse CCA and related problems. In the main reduction below, we use a supervised sparse recovery primitive as a black-box hardness assumption; Appendix G discusses its relationship to standard sparse CCA reductions and the scalar-response caveat.

*Hypothesis* 3.2 (Samplable Supervised Sparse Recovery Primitive). There exist parameters $(d_Z, s)$ with $d_Z = \text{poly}(m)$ and $s = \Theta(\kappa)$, a randomized polynomial-time sampler $\mathcal{R}$, and a sign-invariant population predictive score $\Phi_P(u) = \Phi_P(-u)$ defined for unit vectors $u \in \mathbb{R}^{d_Z}$

and normalized so that $0 \leq \Phi_P(u) \leq 1$, such that, given an average-case input $A$, $\mathcal{R}(A)$ outputs i.i.d. samples $(Z_t, Y_t) \in \mathbb{R}^{d_Z} \times \mathbb{R}$ satisfying the following properties:

(i) under the null case, the samples contain no sparse predictive direction with nontrivial population score;

(ii) under the planted case, there exists an unknown $s$-sparse unit vector $v \in \mathbb{R}^{d_Z}$ whose support encodes the planted structure, the conditional law of $Y$ given $Z$ depends on $Z$ only through $v^\top Z$, and $v$ is the unique sparse predictive direction in the sense that for some constants $c_{\text{prim}} > 0$ and $\delta_{\text{prim}} > 0$,

$$\Phi_P(v) \geq \sup_{\substack{u \in \mathbb{S}^{d_Z-1} \\ \text{dist}(\text{span}(u), \text{span}(v)) \geq \delta_{\text{prim}}}} \Phi_P(u) + c_{\text{prim}}.$$

Moreover, the signal scale

$$s_{\text{cond}} := \left( \mathbb{E}[(\mathbb{E}[Y \mid v^\top Z])^2] \right)^{1/2}$$

is bounded below by a positive constant;

(iii) $\Phi_P(u)$ admits empirical estimates with polynomial-sample uniform concentration over the candidate class used by exhaustive search, and the samples have uniformly bounded moments sufficient for the wrapper covariance statistics to concentrate;

(iv) any polynomial-time algorithm that, from polynomially many samples, outputs a vector with constant overlap with $v$ yields a polynomial-time solver for the underlying average-case sparse-recovery problem;

(v) the underlying average-case sparse-recovery problem is computationally hard in this parameter regime: no randomized polynomial-time algorithm solves it with constant success probability.

**Black-box status and learner access.** Hypothesis 3.2 is a conditional primitive. It is motivated by average-case sparse-recovery and sparse-CCA reductions, but we do not claim that it follows directly from existing Planted-Clique-to-sparse-CCA results. In particular, it is not the exposed scalar covariance model $\text{Cov}(Z, Y) = \lambda v$, which can be easy by empirical covariance estimation in some regimes. The empirical version of the score $\Phi_P$ is part of the primitive interface used by the exhaustive-search learner. Equivalently, the primitive specifies a statistically estimable predictive criterion whose population maximizer is the hidden sparse direction. The reduction in Section 3.2 does not assume access to the hidden support or hidden direction; it only uses samples generated by the primitive and the empirical score estimator guaranteed by Hypothesis 3.2(iii). Therefore the hardness statement should be interpreted as a black-box transfer result for any supervised sparse-recovery task satisfying this interface.

## 3.2. Embedding into Multi-Environment SC Instances

We wrap the supervised sparse primitive into an SC instance. The constraint is that the resulting instance must satisfy Definition 2.1, in particular the invariance condition (C1).

*Construction* 3.3 (Samplable Hard SC Family). *Given a primitive input $A$, independently run $\mathcal{R}(A)$ in each environment $e$ to obtain i.i.d. samples $(Z_t^{(e)}, Y_t^{(e)}) \in \mathbb{R}^{d_Z} \times \mathbb{R}$. In the planted case, let $v$ be the hidden $s$-sparse predictive direction. Set $d_{\mathrm{sp}} = 1$, $d = d_Z + 1$, and*

$$X^{(e,t)} = (Z_t^{(e)}, W^{(e,t)}), \qquad W^{(e,t)} = \mu^{(e)} Y_t^{(e)} + \eta^{(e,t)},$$

*where $\eta^{(e,t)} \sim \mathcal{N}(0,1)$ and $\mu^{(e)}$ is environment-dependent. Let*

$$V_{\mathrm{inv}} = \mathrm{span}\{(v, 0)\}, \qquad V_{\mathrm{sp}} = V_{\mathrm{inv}}^{\perp}.$$

*The construction is samplable from $A$: it uses only independent calls to $\mathcal{R}(A)$ and public Gaussian noise, never the hidden direction or support.*

**Lemma 3.4** (Valid SC Instance). *Construction 3.3 defines an SC instance in the sense of Definition 2.1. In the planted case, the conditional law of $Y$ given $P_{V_{\mathrm{inv}}} X$ is identical across environments because, by Hypothesis 3.2, the conditional law of $Y$ given $Z$ depends on $Z$ only through $v^\top Z$. The spurious block $W^{(e)} = \mu^{(e)} Y + \eta^{(e)}$ creates environment-dependent variation in $X_{\mathrm{sp}} \mid X_{\mathrm{inv}}$ whenever the shifts are not all equal. Moreover, if*

$$\Delta_\mu := \max_{e,e'} |\mu^{(e)} - \mu^{(e')}|,$$

*then in the one-dimensional spurious block used here,*

$$\gamma = \Theta(s_{\mathrm{cond}} \Delta_\mu),$$

*where $s_{\mathrm{cond}} = (\mathbb{E}[(\mathbb{E}[Y \mid v^\top Z])^2])^{1/2}$ is the signal scale in Hypothesis 3.2. Thus constant-size shifts give constant separation whenever $s_{\mathrm{cond}} = \Theta(1)$.*

**Invariance–predictivity score over one-dimensional subspaces.** For a candidate one-dimensional subspace $V \in \mathrm{Gr}(1,d)$, let $u_V = (u_Z, a)$ be any unit vector spanning $V$, where $u_Z \in \mathbb{R}^{d_Z}$ is the primitive block and $a \in \mathbb{R}$ is the spurious block. If $u_Z \neq 0$, write $\bar{u}_Z = u_Z / \|u_Z\|$; if $u_Z = 0$, set the predictive score below to zero. We define the predictive component inherited from the supervised sparse primitive by

$$A(V) := \begin{cases} \|u_Z\|^2 \Phi_P(\bar{u}_Z), & u_Z \neq 0, \\ 0, & u_Z = 0. \end{cases}$$

To penalize environment-dependent spurious components, define the environment-wise association statistic

$$\theta_V^{(e)} := \mathrm{Cov}_e(Y, u_V^\top X)$$

and

$$T(V) := \max_{e \neq e'} \left( \theta_V^{(e)} - \theta_V^{(e')} \right)^2.$$

The covariance statistic is used only to detect the environment-dependent spurious block in the wrapper; the predictive hardness is inherited from the black-box primitive through $\Phi_P$. We combine the two terms into

$$S(V) := A(V) - \lambda T(V),$$

where $\lambda > 0$ is chosen large enough that non-invariant spurious directions are penalized more than any gain in predictivity. In Construction 3.3, many directions can be invariant but non-predictive; the score $S(V)$ combines invariance with predictive power to rule out such directions.

**Lemma 3.5** (Uniform Score Margin). *Consider Construction 3.3 in the planted case. Let*

$$\Delta_\mu := \max_{e,e'} |\mu^{(e)} - \mu^{(e')}| > 0, \qquad \sigma_Y^2 := \mathrm{Var}(Y),$$

*and assume $\sigma_Y^2$ is bounded above and below by positive constants. Suppose the primitive score satisfies Hypothesis 3.2 with margin $c_{\mathrm{prim}} > 0$. Then for every fixed constant recovery radius $\delta \in (0,1)$, there exist constants $\lambda > 0$ and $c_{\mathrm{mar}} > 0$, depending only on $\delta, c_{\mathrm{prim}}, \Delta_\mu, \sigma_Y^2$ and the primitive moment bounds, such that*

$$S(V_{\mathrm{inv}}) \geq \sup_{\substack{V \in \mathrm{Gr}(1,d) \\ \mathrm{dist}(V, V_{\mathrm{inv}}) \geq \delta}} S(V) + c_{\mathrm{mar}}.$$

*In particular, taking $\delta = 0.1$ gives the margin used in Lemma 3.6.*

## 3.3. Statistical Learnability vs. Computational Hardness

Given Lemma 3.5, an exponential-time search over an $\epsilon$-net of $\mathrm{Gr}(1,d)$ can recover the invariant one-dimensional subspace by maximizing an empirical version of $S(V)$. The required sample size follows from uniform concentration over the finite net and environment pairs.

**Lemma 3.6** (Statistical Learnability). *There exists an exponential-time algorithm that recovers $V_{\mathrm{inv}}$ in Construction 3.3 with a polynomial number of samples $N = \tilde{O}(\mathrm{poly}(m)/c_{\mathrm{mar}}^2)$.*

The proof follows by uniform concentration over an $\epsilon$-net of $\mathrm{Gr}(1,d)$ and is given in Appendix F.8.

To obtain hardness, we show that constant-accuracy recovery of $V_{\mathrm{inv}}$ yields constant-overlap recovery of the hidden sparse direction in the supervised sparse primitive. Any algorithm that outputs $\hat{V}$ with $\mathrm{dist}(\hat{V}, V_{\mathrm{inv}}) \leq \delta_0$ induces a constant-accuracy recovery of the direction $v$ in the primitive, which by Hypothesis 3.2(iv) yields a solver for the underlying average-case sparse-recovery problem.

**Lemma 3.7** (Reduction Correctness)**.** *There exists an absolute constant* $\delta_0 \in (0, 1/2)$ *such that if an algorithm outputs* $\hat{V}$ *with* $\mathrm{dist}(\hat{V}, V_{\mathrm{inv}}) = \|P_{\hat{V}} - P_{V_{\mathrm{inv}}}\|_F \leq \delta_0$ *on Construction 3.3, then one obtains a polynomial-time solver for the underlying average-case sparse-recovery problem with probability* $\geq 2/3$.

The proof is by projecting the recovered one-dimensional subspace onto the primitive block and invoking the decoder in Hypothesis 3.2; see Appendix F.9.

Any approximation or sampling discrepancy included in a concrete instantiation of the primitive only changes success probabilities by the corresponding total-variation amount, so the constant-success decoding implication remains valid whenever the primitive provides such a guarantee.

**Theorem 3.8** (Computational Hardness)**.** *Assume the black-box samplable supervised sparse recovery primitive in Hypothesis 3.2. There exist samplable SC instances with tunable separation, in particular with* $\gamma = \Theta(1)$ *when the signal scale satisfies* $s_{\mathrm{cond}} = \Theta(1)$ *and the environment shifts are constant-size, such that:*

*(a) $V_{\mathrm{inv}}$ is* invariant-predictive identifiable*: it is the unique maximizer of $S(V)$ over $\mathrm{Gr}(1, d)$ (and in particular achieves $T(V) = 0$);*
*(b) exhaustive search recovers $V_{\mathrm{inv}}$ with polynomially many samples;*
*(c) any polynomial-time algorithm achieving $\mathrm{dist}(\hat{V}, V_{\mathrm{inv}}) \leq \delta_0$ at this sample size would yield a polynomial-time solver for the underlying average-case sparse-recovery problem, and hence contradict Hypothesis 3.2.*

**Theorem 3.9** (Computational–Statistical Gap)**.** *Under the black-box samplable supervised sparse recovery primitive in Hypothesis 3.2, there exists a polynomial sample size $N_{\mathrm{stat}}$ at which exhaustive search succeeds, whereas any polynomial-time algorithm achieving constant-accuracy recovery of $V_{\mathrm{inv}}$ on the hard family would contradict the hardness clause of the primitive. Here constant-accuracy means $\mathrm{dist}(\hat{V}, V_{\mathrm{inv}}) \leq \delta_0$ for the constant $\delta_0$ in Lemma 3.7.*

*Remark* 3.10 (Few environments)**.** Hardness does not rely on many environments: the construction already works with a constant number of environments (e.g., $|\mathcal{E}| = 2$) provided $\mu^{(1)} \neq \mu^{(2)}$ so that $T(V)$ detects spurious components.

# 4. Statistical Foundations

This section characterizes when $V_{\mathrm{inv}}$ is identifiable and how sample complexity scales. The key quantity is *environment diversity* (Definition 2.3): insufficient diversity makes recovery statistically impossible, while sufficient diversity yields well-posed recovery with sharp finite-sample transitions.

## 4.1. Identifiability via Diversity

**Definition 4.1** (Invariance Gap)**.** For $V \in \mathrm{Gr}(k, d)$, define $X_V := P_V X$ and

$$\mathcal{I}_{\mathrm{gap}}(V) := \max_{e \neq e'} \mathbb{E}_{X_V}[D_{\mathrm{KL}}(\mathbb{P}_e(Y \mid X_V) \,\|\, \mathbb{P}_{e'}(Y \mid X_V))],$$

where the expectation is taken over $X_V$ under the marginal distribution induced by $\mathbb{P}_e$.

**Definition 4.2** (Criterion-identifiability)**.** We say that $V_{\mathrm{inv}}$ is *identifiable by the invariance-gap criterion* over a model class $\mathcal{P}$ if, for the population family $\{\mathbb{P}_e\}_{e \in \mathcal{E}}$, every $k$-dimensional subspace $V$ satisfying

$$\mathcal{I}_{\mathrm{gap}}(V) = 0$$

must equal $V_{\mathrm{inv}}$. Equivalently, for every $\epsilon > 0$, compactness and continuity imply the existence of a population separation

$$\inf_{\mathrm{dist}(V, V_{\mathrm{inv}}) \geq \epsilon} \mathcal{I}_{\mathrm{gap}}(V) > 0.$$

This is a criterion-level notion: it asks whether the population invariance objective uniquely specifies the invariant subspace, rather than whether a particular finite-sample algorithm succeeds.

By (C1), $\mathcal{I}_{\mathrm{gap}}(V_{\mathrm{inv}}) = 0$. Intuitively, if $V$ contains only invariant information, then the predictive mechanism $Y \mid X_V$ should not change across environments; if $V$ mixes in spurious directions, conditioning on $X_V$ can induce environment-dependent behavior.

**Theorem 4.3** (Identifiability by the Invariance-Gap Criterion)**.** *Under Assumption 2.2, $V_{\mathrm{inv}}$ is identifiable by the invariance-gap criterion in the sense of Definition 4.2 if and only if it is the unique $k$-dimensional subspace satisfying $\mathcal{I}_{\mathrm{gap}}(V) = 0$.*

Proofs for this section are deferred to Appendix F.

**A structured label-induced shift regime.** In many spurious-correlation settings, environments affect *how spurious features correlate with the label* rather than changing the $Y \mid X_{\mathrm{inv}}$ mechanism. We formalize this next.

**Proposition 4.4** (Diversity Condition (Label-Induced Shift))**.** *Assume a linear-Gaussian SC instance where:*

*1. $Y = \langle w^*, X_{\mathrm{inv}} \rangle + \epsilon$ with $w^* \in V_{\mathrm{inv}}$, $\|w^*\| = 1$, and $\epsilon \sim \mathcal{N}(0, \sigma_\epsilon^2)$ independent of $X_{\mathrm{inv}}$;*
*2. $X_{\mathrm{inv}} \sim \mathcal{N}(0, I_k)$ in every environment;*
*3. environments differ only through a label-induced spurious shift:*

$$X_{\mathrm{sp}}^{(e)} = \mu_{\mathrm{sp}}^{(e)} Y + \eta^{(e)}, \qquad \eta^{(e)} \sim \mathcal{N}(0, I_{d-k}),$$

*independently of $(X_{\mathrm{inv}}, Y)$.*

*Let*

$$\mathcal{F} := \sum_{e < e'} (\mu_{\mathrm{sp}}^{(e)} - \mu_{\mathrm{sp}}^{(e')})(\mu_{\mathrm{sp}}^{(e)} - \mu_{\mathrm{sp}}^{(e')})^{\top} \in \mathbb{R}^{(d-k) \times (d-k)}.$$

*Then, generically (i.e., outside the algebraic non-cancellation degeneracy set characterized in Appendix F.3), $V_{\mathrm{inv}}$ is identifiable (i.e., it is the unique $k$-dimensional subspace with $\mathcal{I}_{\mathrm{gap}}(V) = 0$) whenever $\mathrm{rank}(\mathcal{F}) = d - k$.*

*Remark* 4.5 (Number of environments required by the full-rank condition). Since $\mathrm{rank}(\mathcal{F}) \leq |\mathcal{E}| - 1$, the sufficient condition $\mathrm{rank}(\mathcal{F}) = d - k$ requires $|\mathcal{E}| \geq d - k + 1$. Thus Proposition 4.4 gives a full-dimensional sufficient condition. With fewer environments, identifiability can still hold under lower-dimensional or structured spurious variation, but not through this full-rank criterion.

*Remark* 4.6 (Generic non-cancellation). The rank condition ensures that every nonzero spurious component is exposed by some environment difference. In rotated subspace models, exact Gaussian conditional-law cancellations can occur only on an algebraic degeneracy set; Appendix F.3 gives the formal generic non-cancellation argument.

**Corollary 4.7** (Zero Diversity Implies Unidentifiability (in the label-induced model)). *Under the setting of Proposition 4.4, if $\mu_{\mathrm{sp}}^{(e)}$ is identical across all environments $e \in \mathcal{E}$, then $V_{\mathrm{inv}}$ is unidentifiable regardless of sample size.*

*Remark* 4.8 (On $\gamma = 0$ versus "no variation of $\mu_{\mathrm{sp}}^{(e)}$"). In the label-induced model, $\mathbb{E}_e[X_{\mathrm{sp}} \mid X_{\mathrm{inv}}] = \mu_{\mathrm{sp}}^{(e)} \mathbb{E}[Y \mid X_{\mathrm{inv}}] = \mu_{\mathrm{sp}}^{(e)} \langle w^*, X_{\mathrm{inv}} \rangle$. Thus, if $\mu_{\mathrm{sp}}^{(e)}$ does not vary across environments then $\gamma = 0$ by Definition 2.3. The converse ("$\gamma = 0$ implies $\mu_{\mathrm{sp}}^{(e)}$ identical") can fail in degenerate cases where $\mathbb{E}[Y \mid X_{\mathrm{inv}}] \equiv 0$. Our standing linear-Gaussian assumption with $\|w^*\| = 1$ rules out such degeneracy.

### 4.2. Minimax Rates

We next quantify the best possible estimation error with unlimited computation. The term $k(d - k)$ is the intrinsic degrees of freedom of a $k$-dimensional subspace in $\mathbb{R}^d$, while the factor $n|\mathcal{E}|$ reflects the effective total sample size across environments in the balanced case.

**Definition 4.9** (Spurious Complexity). For $\epsilon > 0$, define the $\epsilon$-confusing set

$$\mathcal{V}_{\mathrm{conf}}(\epsilon) := \{V \in \mathrm{Gr}(k, d) : \mathcal{I}_{\mathrm{gap}}(V) \leq \epsilon\},$$

and its metric entropy

$$\mathcal{C}_{\mathrm{sp}}(\mathcal{E}, \epsilon) := \log \mathcal{N}(\mathcal{V}_{\mathrm{conf}}(\epsilon), \epsilon),$$

where $\mathcal{N}(\cdot, \epsilon)$ denotes the $\epsilon$-covering number under the subspace distance $\mathrm{dist}(\cdot, \cdot)$.

The quantity $\mathcal{C}_{\mathrm{sp}}$ captures how many subspaces appear approximately invariant: high diversity shrinks $\mathcal{V}_{\mathrm{conf}}(\epsilon)$ to a

neighborhood of $V_{\mathrm{inv}}$, while low diversity can create many near-invariant "confusers."

**Assumption 4.10** (Regularity Conditions for Minimax Rates).

  (i) $\sigma_{\min}^2 I \preceq \Sigma^{(e)} \preceq \sigma_{\max}^2 I$ for all $e$ with condition number $\kappa_{\Sigma} := \sigma_{\max}^2 / \sigma_{\min}^2$;
 (ii) the linear-Gaussian structural model holds with $\|w^*\| = 1$ and $\mathrm{SNR} := 1/\sigma_{\epsilon}^2 \in [\underline{\mathrm{SNR}}, \overline{\mathrm{SNR}}]$;
(iii) $\gamma \geq \gamma_0 > 0$ and the invariance gap has nondegenerate local quadratic curvature: for some constants $r_0, c_{\mathrm{id}}, C_{\mathrm{id}} > 0$,

$$c_{\mathrm{id}} \gamma_0^2 \, \mathrm{dist}(V, V_{\mathrm{inv}})^2 \leq \mathcal{I}_{\mathrm{gap}}(V) \leq C_{\mathrm{id}} \, \mathrm{dist}(V, V_{\mathrm{inv}})^2$$

   whenever $\mathrm{dist}(V, V_{\mathrm{inv}}) \leq r_0$, and $\mathcal{I}_{\mathrm{gap}}(V)$ is bounded away from zero outside this neighborhood;
(iv) the local Gaussian likelihood in a Grassmannian chart around $V_{\mathrm{inv}}$ satisfies standard LAN regularity, with Fisher information bounded above and below by constants depending only on $(\kappa_{\Sigma}, \mathrm{SNR}, \gamma_0, c_{\mathrm{id}}, C_{\mathrm{id}})$.

**Interpretation of the minimax theorem.** The next result is a regular local minimax statement. Its assumptions explicitly include local quadratic curvature of the invariance gap and LAN-type Gaussian regularity. Thus the theorem should not be read as deriving these regularity properties in full generality. Rather, it states that once environment diversity yields a locally well-conditioned invariant subspace problem, the optimal statistical rate is the regular parametric Grassmannian rate with dimension $k(d - k)$ and total sample size $n|\mathcal{E}|$. The label-induced shift model in Section 4.3 then gives one concrete setting in which the local curvature scales with diversity.

**Theorem 4.11** (Regular Local Minimax Estimation Risk). *Under Assumptions 2.2 and 4.10 with $n$ per environment (balanced):*

**Lower bound.** *For any estimator $\hat{V}$,*

$$\sup_{\mathbb{P}} \mathbb{E}[\mathrm{dist}(\hat{V}, V_{\mathrm{inv}})^2] \geq c_{\mathrm{low}}(\kappa_{\Sigma}, \mathrm{SNR}, c_{\mathrm{id}}, \gamma_0) \cdot \frac{k(d - k)}{n|\mathcal{E}|}.$$

**Upper bound.** *Moreover, there exists an (inefficient) estimator such that*

$$\sup_{\mathbb{P}} \mathbb{E}\big[\mathrm{dist}(\hat{V}, V_{\mathrm{inv}})^2\big] \leq C(\kappa_{\Sigma}, \mathrm{SNR}, \gamma_0, c_{\mathrm{id}}, C_{\mathrm{id}}) \cdot \frac{1}{n|\mathcal{E}|}$$
$$\times \Big(k(d - k) + \mathcal{C}_{\mathrm{sp}}\big(\mathcal{E}, c_1/\sqrt{n|\mathcal{E}|}\big)\Big),$$

*where $c_1 > 0$ is an absolute constant corresponding to the estimation resolution. In particular, when $\gamma = \Theta(1)$ and $\mathcal{C}_{\mathrm{sp}} = O(k(d-k))$, the minimax risk is $\Theta(k(d-k)/(n|\mathcal{E}|))$.*

**Corollary 4.12** (Sample Complexity for Target Error). *Under Theorem 4.11, to achieve $\mathbb{E}[\mathrm{dist}(\hat{V}, V_{\mathrm{inv}})^2] \leq \epsilon^2$, it*

*suffices (up to log factors) that*

$$n \geq \tilde{\Theta}\left(\frac{k(d-k)}{|\mathcal{E}|\epsilon^2}\right),$$

*with constants depending on* $(\kappa_\Sigma, \mathrm{SNR})$ *and on diversity through the local curvature/identifiability of* $\mathcal{I}_{\mathrm{gap}}$.

*Remark* 4.13 (Scope of $\gamma$-dependence). The minimax rate above is stated under $\gamma \geq \gamma_0 > 0$; in general $\gamma$ affects constants via identifiability/curvature. An explicit $1/\gamma^2$ scaling is available under additional label-induced shift structure.

### 4.3. Phase Transition and Explicit $\gamma$-Scaling

To make the role of diversity explicit, we specialize to the label-induced shift regime (Proposition 4.4). In this setting, diversity controls the smallest eigenvalue of the environment-difference matrix $\mathcal{F}$ and hence the local curvature of invariance objectives around $V_{\mathrm{inv}}$.

**Assumption 4.14** (Label-Induced Shift Model for Phase Transition). In addition to Assumption 4.10:

(i) environments differ through a label-induced spurious shift as in Proposition 4.4, and the induced covariance matrices are uniformly well-conditioned across environments as required by Assumption 4.10;

(ii) $\lambda_{\min}(\mathcal{F}) \geq c_{\mathcal{F}}\gamma^2$ for some $c_{\mathcal{F}} > 0$.

**Directional diversity in the phase transition.** The explicit $1/\gamma^2$ scaling below uses Assumption 4.14, which links the coordinate-level diagnostic $\gamma$ to the spectral directional diversity $\lambda_{\min}(\mathcal{F})$. More generally, the same argument gives rates controlled by $1/\lambda_{\min}(\mathcal{F})$; the displayed $1/\gamma^2$ form applies when $\lambda_{\min}(\mathcal{F}) \geq c_{\mathcal{F}}\gamma^2$.

**Theorem 4.15** (Phase Transition). *Under Assumptions 2.2, 4.10, and 4.14, define the critical sample size (per environment)*

$$n^* := C_0(\kappa_\Sigma, \mathrm{SNR}, c_{\mathcal{F}}) \cdot \frac{k(d-k)}{|\mathcal{E}|\gamma^2}.$$

***Super-critical*** *($n > 2n^*$): there exists an estimator with*

$$\mathbb{E}\left[\mathrm{dist}(\hat{V}, V_{\mathrm{inv}})^2\right] \leq C_1 \frac{k(d-k)}{n|\mathcal{E}|\gamma^2}.$$

*In particular, when* $\gamma = \Theta(1)$*, this reduces to the rate* $O(k(d-k)/(n|\mathcal{E}|))$*.* ***Sub-critical*** *($n < n^*/2$): any estimator satisfies*

$$\sup_{\mathbb{P}} \mathbb{E}\left[\mathrm{dist}(\hat{V}, V_{\mathrm{inv}})^2\right] \geq c_{\mathrm{sub}},$$

*where* $c_{\mathrm{sub}} > 0$ *is a constant depending on* $(\kappa_\Sigma, \mathrm{SNR}, c_{\mathcal{F}})$.

*Remark* 4.16 (Constant dependencies). The constants depend on the covariance condition number $\kappa_\Sigma$, the signal-to-noise ratio, and the diversity structure constant $c_{\mathcal{F}}$. The refined estimation error contains the factor $1/\gamma^2$; the rate $k(d-k)/(n|\mathcal{E}|)$ is recovered in the constant-diversity regime $\gamma = \Theta(1)$.

## 5. Tractability Conditions

The hardness result is worst-case. In many practical settings, additional structure gives efficient certificates for recovering $V_{\mathrm{inv}}$. We summarize four sufficient regimes.

**Theorem 5.1** (Tractability templates, informal). *Under Assumptions 2.2 and 4.10, polynomial-time or mildly superpolynomial estimators achieve*

$$\mathbb{E}\left[\mathrm{dist}(\hat{V}, V_{\mathrm{inv}})^2\right] = \tilde{O}\left(\frac{k(d-k)}{n|\mathcal{E}|}\right)$$

*whenever one of the following structures holds:*

1. ***Sparse invariant structure:*** $V_{\mathrm{inv}}$ *has a sparse basis, reducing recovery to support search.*
2. ***Spectral separation:*** *an efficiently computable cross-environment contrast matrix has an eigengap separating invariant and spurious directions.*
3. ***Low-order moment identifiability:*** *spurious directions induce detectable environment variation in correlations or moments, while invariant directions remain stable.*
4. ***Environment-wise uncorrelatedness:*** *invariant and spurious features are uncorrelated within each environment, enabling coordinate-wise tests.*

A practical diagnostic is to compute per-feature correlation ranges

$$v_j = \max_e \hat{\rho}_j^{(e)} - \min_e \hat{\rho}_j^{(e)}, \qquad \hat{\rho}_j^{(e)} = \mathrm{Corr}(X_j^{(e)}, Y^{(e)}),$$

or the representation-space proxy $\hat{\gamma}_{\mathrm{repr}}$. Large separation between stable predictive coordinates and varying spurious coordinates suggests that simple screening, spectral methods, or group-aware objectives should be effective. Detailed algorithms and checks are given in Appendix I.

### 5.1. Connection to Real Benchmarks

Our tractability conditions provide a compact interpretation of prior benchmark behavior: datasets with low-dimensional, strongly shifting spurious signals (e.g., ColoredMNIST/CelebA/Waterbirds) approximately satisfy (T3) and favor moment- or group-aware methods, whereas high-dimensional and subtle shifts (e.g., DomainBed-style) can weaken (T2)–(T4). More discussion is in Appendix H.

## 6. Experiments

We empirically study three aspects of our framework: (i) compute–sample gaps on synthetic hard instances; (ii) phase transitions and diversity-dependent thresholds; (iii) behavior on standard spurious-correlation benchmarks. Details and ablations are in Appendix H.

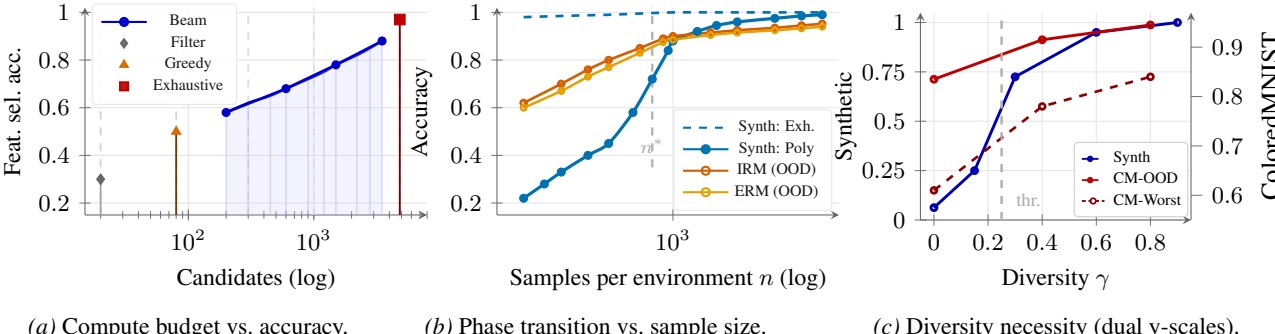

*(a)* Compute budget vs. accuracy.  *(b)* Phase transition vs. sample size.  *(c)* Diversity necessity (dual y-scales).

*Figure 2.* Empirical signatures: (a) compute–accuracy tradeoff; (b) phase transition vs. sample size; (c) effect of diversity.

*Table 1.* ColoredMNIST ($\gamma = 0.4$): OOD accuracy at fixed total training size $N$ (mean $\pm$ std over 5 seeds). The gap shrinks with more samples, consistent with phase-transition behavior.

| $N$ | Method | OOD Acc | Gap vs Oracle |
|---|---|---|---|
| | Oracle (gray) | .820 $\pm$ .012 | — |
| 500 | Adv-Color | .745 $\pm$ .018 | 0.075 |
| | IRM | .720 $\pm$ .022 | 0.100 |
| | ERM | .690 $\pm$ .025 | 0.130 |
| | Oracle (gray) | .930 $\pm$ .008 | — |
| 2000 | Adv-Color | .915 $\pm$ .010 | 0.015 |
| | IRM | .910 $\pm$ .009 | 0.020 |
| | ERM | .900 $\pm$ .012 | 0.030 |

### 6.1. Experimental Settings

**Synthetic data.** We make axis-aligned instances inspired by the hardness mechanisms in Section 3, where invariant coordinates are hidden among spurious "confusers" whose cross-environment statistics partially mimic invariance. We report *feature selection accuracy* (fraction of true invariant coordinates among the selected top-$k$). We compare exhaustive subset search (exponential time) against polynomial-time baselines including invariance screening (Section 5) and other efficient heuristics (Appendix H). Unless stated otherwise we use $d = 20$, $k = 4$, $|\mathcal{E}| = 4$ and vary $n$ or $\gamma$. These instances are not intended to implement the planted-clique reduction; rather, they probe compute–sample trade-offs in regimes that create many near-invariant alternatives.

**Real datasets.** We evaluate on ColoredMNIST, CelebA (Blond_Hair with spurious Male), and Waterbirds (bird type with spurious background). For ColoredMNIST we explicitly control diversity by varying the difference in color-label flip probabilities across training environments, which serves as a practical proxy for $\gamma$ in Definition 2.3.

### 6.2. Gap, Phase Transition, and Diversity

**(1) Computational gap.** On synthetic hard instances, Fig. 1 shows a large sample-complexity gap: exhaustive

search achieves a fixed target accuracy with up to $27\times$ fewer samples than the best polynomial-time method. Holding the dataset fixed, Fig. 2a further shows a monotone compute–accuracy frontier: moving from simple filtering ($O(d)$) to greedy ($O(kd)$), to beam search, and finally to exhaustive evaluation consistently improves recovery.

On real data, Table 1 exhibits an analogous small-sample separation between an oracle invariant pipeline (grayscale) and learned invariance-promoting methods: at $N = 500$ IRM trails the oracle by 10 points, while the gap shrinks at $N = 2000$, consistent with increased data easing the learning problem. Since the oracle uses grayscale preprocessing (different input), this gap should be interpreted cautiously: it suggests the difficulty of *discovering* invariant structure from finite multi-environment data rather than proving computational hardness for deep networks. Adv-Color partially controls for input differences by using the same colored input while explicitly suppressing color information.

**(2) Phase transitions.** Fig. 2b shows phase-transition behavior: on synthetic instances, polynomial-time methods transition sharply from failure to success around a critical sample size, while exhaustive search stays near-optimal across the range. On ColoredMNIST, the same figure shows a smoother but qualitatively similar improvement with $n$, and Table 1 confirms that the small-sample regime is where invariant learning is most brittle and additional samples quickly reduce the gap. Additional training dynamics are deferred to Appendix (Fig. 3).

**(3) Diversity is necessary.** Fig. 2c visualizes diversity necessity: on synthetic data, performance is poor when $\gamma = 0$ and increases sharply once environments become informative; on ColoredMNIST, both OOD and worst-group accuracy improve as diversity increases. Table 2 extends this pattern across datasets: moving from the operational low-diversity/high-bias setting to the operational high-diversity/low-bias setting improves worst-group accuracy on CelebA and Waterbirds and improves OOD ac-

*Table 2.* Real benchmark results (mean $\pm$ std over 5 seeds). For real benchmarks, "low/high diversity" denotes an operational proxy based on the strength and variability of the spurious bias; this proxy is not identical to the formal $\gamma$ but reflects how informative the environments are for separating invariant and spurious signals.

| Dataset | Method | Low Diversity | | High Diversity | |
|---|---|---|---|---|---|
| | | Overall | Worst | Overall | Worst |
| CMNIST | ERM | .82±.01 | .54±.02 | .92±.01 | .76±.02 |
| | IRM | .84±.01 | .61±.02 | **.95±.01** | **.84±.01** |
| CelebA | ERM | .90±.01 | .76±.02 | .93±.01 | .89±.01 |
| | IRM | .89±.01 | .82±.02 | .92±.01 | .90±.01 |
| | GDRO | .89±.01 | **.85±.01** | .92±.01 | **.91±.01** |
| Wbirds | ERM | .85±.02 | .68±.03 | .87±.01 | .84±.02 |
| | IRM | .84±.02 | .74±.02 | .87±.01 | .86±.02 |
| | GDRO | .83±.02 | **.79±.02** | .87±.01 | **.87±.01** |

CMNIST: Low=$\gamma = 0$, High=$\gamma = 0.8$. CelebA/Wbirds use bias strength as an operational proxy: Low=shared spurious bias 0.9, High=reduced shared bias 0.0.

*Table 3.* Representation-space diversity ($\hat{\gamma}_{\mathrm{repr}}$) computed from ERM penultimate-layer representations, versus downstream performance (IRM test accuracy; mean $\pm$ std over 5 seeds).

| Dataset | Setting | $\hat{\gamma}_{\mathrm{repr}}$ | Worst-Grp | Overall |
|---|---|---|---|---|
| CMNIST | Low div. | .12 ± .03 | .61 ± .02 | .84 ± .01 |
| | High div. | .45 ± .04 | .84 ± .01 | .95 ± .01 |
| CelebA | High bias | .18 ± .04 | .82 ± .02 | .89 ± .01 |
| | Low bias | .38 ± .05 | .90 ± .01 | .92 ± .01 |
| Wbirds | High bias | .22 ± .05 | .74 ± .02 | .84 ± .02 |
| | Low bias | .42 ± .06 | .86 ± .02 | .87 ± .01 |

curacy on ColoredMNIST. Finally, Table 3 shows that a representation-space proxy $\hat{\gamma}_{\mathrm{repr}}$ computed on ERM features correlates with worst-group performance, supporting diversity estimates as a practical diagnostic.

### 6.3. Tractability Validation on Synthetic Data

Table 4 connects Section 5 to empirical performance by varying structural factor at a time. Increasing diversity ($\gamma = 0.1 \rightarrow 0.6$) yields the largest gain, consistent with the identifiability/curvature role of $\gamma$ (Section 4). Increasing sample size ($n = 200 \rightarrow 2000$) improves performance but with diminishing returns once diversity is high, matching phase-transition intuition. Reducing $k$ (a proxy for sparsity / effective search-space size) improves recovery, aligning with (T1). Increasing SNR helps but does not replace diversity, illustrating noise and diversity are distinct bottlenecks.

### 6.4. Practical Guidance and Diagnostics

Compute a diversity proxy $\hat{\gamma}_{\mathrm{repr}}$ in representation space (Table 3) by applying Definition 2.5 to learned features. If it is small, collecting more diverse environments is typically

*Table 4.* Tractability conditions validation on synthetic data. Values are feature selection accuracy (mean $\pm$ std over 10 seeds).

| Condition | Baseline | Improved | $\Delta$ Acc |
|---|---|---|---|
| Diversity: $\gamma = 0.1 \rightarrow 0.6$ | .20 ± .04 | 1.0 ± .00 | +.80 |
| Samples: $n = 200 \rightarrow 2000$ | .70 ± .06 | 1.0 ± .00 | +.30 |
| Sparsity: $k = 8 \rightarrow k = 3$ | .73 ± .05 | 1.0 ± .00 | +.27 |
| Signal: SNR $= 1 \rightarrow 10$ | .75 ± .05 | .93 ± .03 | +.18 |

more effective than collecting more samples from similar environments. When performance improves monotonically with compute at fixed data (Fig. 2a), the instance behaves like a hard regime where efficient objectives may struggle.

**Related Work.** We build on spurious-correlation and invariant learning benchmarks (e.g., IRM/REx and Group-DRO) and on average-case computational lower bounds for sparse recovery, including Planted-Clique-motivated samplable reductions. A detailed discussion and references are deferred to Appendix C.

## 7. Conclusion

We establish a conditional computational–statistical gap for spurious correlation removal: under a black-box samplable supervised sparse recovery primitive, there exist samplable multi-environment instances where the predictive invariant subspace is identifiable as the unique maximizer of an invariance–predictivity population score and learnable with polynomial samples by exhaustive search, yet any polynomial-time constant-accuracy recovery algorithm at comparable sample sizes would contradict the primitive. We also characterize the role of environment diversity in identifiability and finite-sample behavior, obtaining minimax risk $\Theta(k(d-k)/(n|\mathcal{E}|))$ under sufficient diversity and local Gaussian regularity, and a label-induced phase transition at $n^* \propto k(d-k)/(|\mathcal{E}|\gamma^2)$ with refined estimation error scaling proportional to $1/\gamma^2$. Finally, we give tractability conditions under which efficient methods achieve near-optimal rates. Future work includes sharper gap frameworks and extending lower bounds beyond linear-Gaussian models.

## Acknowledgements

This work was partially supported by the "Pioneer" and "Leading Goose" R&D Program of Zhejiang (Grant No.2025C01037) and the Key R&D Program Project of Hangzhou (Grant No.2024SZD1A03).

## Impact Statement

**Positive.** Our framework helps practitioners diagnose failures in invariant learning by distinguishing statistical limitations (insufficient samples or diversity) from potential

computational barriers (hard problem structure). This can:

- reduce wasted effort on algorithmic improvements when the issue is data diversity;
- guide data collection toward more informative environments rather than simply more samples;
- improve deployment decisions by identifying when learned models may be unreliable under shift.

**Potential negative impacts.** Conditional worst-case hardness results could be misinterpreted as discouraging work on invariant learning or as evidence that such methods are fundamentally flawed. We emphasize:

- many practical problems satisfy tractability conditions (Section 5), explaining why invariant learning often works;
- our hardness results are worst-case conditional constructions, not claims about typical problems.

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

# A. Overview of Appendix

This appendix provides formal proofs, proof refinements, implementation details, extended experiments, and practical guidance supporting the main text.

It is organized as follows:

- Appendix B: notation summary.

- Appendix C: additional related work and positioning.

- Appendix D: formal restatements and logical status of the main theoretical claims.

- Appendix E: status of the supervised sparse primitive and scalar covariance sanity checks.

- Appendix F: complete proofs and proof refinements.

- Appendix G: connection to Planted-Clique-based sparse CCA reductions.

- Appendix H: experimental details and extended results.

- Appendix I: practical diagnostic guide.

**Important logical clarification.** The computational lower bound uses the supervised sparse recovery primitive in Hypothesis 3.2 as a black-box average-case hardness primitive. Existing Planted-Clique-to-sparse-CCA reductions motivate this type of sparse-recovery hardness, but sparse CCA is naturally a two-view/vector-response problem. We therefore do not claim that the supervised scalar-response primitive follows directly from existing sparse CCA reductions. In particular, the primitive should not be interpreted as the exposed scalar covariance model $\mathrm{Cov}(Z, Y) = \lambda v$, which can be easy by empirical covariance estimation in some regimes.

# B. Notation Summary

Table 5 summarizes the main symbols used in the paper and appendix. We separate statistical quantities such as $\gamma$, $\mathcal{I}_{\mathrm{gap}}$, and $\mathcal{C}_{\mathrm{sp}}$ from computational-hardness quantities such as $d_Z$, $s$, $v$, and $\Phi_P$, reflecting the two components of our framework.

# C. Additional Related Work and Positioning

This section expands the literature context and clarifies how our results differ from existing work on invariant learning, identifiability, and computational lower bounds.

### C.1. Invariant learning and identifiability

IRM, REx, GroupDRO, and related methods aim to learn predictors whose label mechanism is stable across environments. Prior theory emphasizes that multi-environment information is useful only when environments are sufficiently diverse; otherwise, invariances can be statistically unidentifiable. Our work is complementary: even when invariant structure is identifiable in a population sense, efficient recovery can still be obstructed under a conditional sparse-recovery hardness primitive.

### C.2. Computational-statistical gaps

Average-case reductions from Planted Clique are a standard tool for demonstrating computational-statistical gaps in high-dimensional statistics, especially sparse PCA and sparse CCA. Our reduction differs in that the target problem is multi-environment invariant subspace recovery. The embedding must preserve the invariance condition and must be samplable without using the hidden sparse direction or its support.

*Table 5.* Notation used throughout the paper and appendix.

| Symbol | Description |
|---|---|
| $d$ | Ambient feature dimension |
| $k$ | Dimension of invariant subspace $V_{\text{inv}}$ |
| $p = d - k$ | Dimension of spurious subspace |
| $q = k(d - k)$ | Local dimension of $\text{Gr}(k, d)$ |
| $\mathcal{E}$ | Set of environments |
| $|\mathcal{E}|$ | Number of environments |
| $n$ | Samples per environment in balanced case |
| $N = n|\mathcal{E}|$ | Total sample size in balanced case |
| $V_{\text{inv}}, V_{\text{sp}}$ | Invariant and spurious subspaces |
| $P_V$ | Orthogonal projector onto $V$ |
| $\text{Gr}(k, d)$ | Grassmannian of $k$-dimensional subspaces in $\mathbb{R}^d$ |
| $\text{dist}(V, V')$ | Subspace distance $\|P_V - P_{V'}\|_F$ |
| $\gamma$ | Environment separation parameter |
| $\hat{\gamma}$ | Empirical diversity proxy |
| $\hat{\gamma}_{\text{repr}}$ | Diversity proxy in representation space |
| $\mathcal{I}_{\text{gap}}(V)$ | Invariance gap functional |
| $\mathcal{C}_{\text{sp}}$ | Confusing-set metric entropy |
| $d_Z, s, c_{\text{prim}}, s_{\text{cond}}$ | Sparse primitive dimension, sparsity, primitive margin, and signal scale |
| $v$ | Hidden sparse direction in primitive |
| $\mu^{(e)}$ | Environment-specific spurious shift |
| $\theta_V^{(e)}$ | Environment-wise covariance score |
| $A(V), T(V), S(V)$ | Predictivity, invariance penalty, combined score |

### C.3. Relation to sparse CCA and scalar supervised primitives

Sparse CCA reductions naturally produce vector-response two-view models. The supervised scalar-response sparse primitive used in the main text is treated as a black-box primitive rather than as a direct consequence of sparse CCA. Appendix G explains the relationship and the limitation of directly projecting sparse CCA to a scalar response.

## D. Formal Restatements and Logical Status

*Hypothesis* D.1 (Formal restatement of Hypothesis 3.2). This is the appendix restatement of the black-box samplable supervised sparse recovery primitive in Hypothesis 3.2. Namely, there exist parameters $(d_Z, s)$, with $d_Z = \text{poly}(m)$ and $s = \Theta(\kappa)$, a randomized polynomial-time sampler $\mathcal{R}$, and a sign-invariant population predictive score $\Phi_P$ such that the null, planted, concentration, decoding, and hardness properties stated in Hypothesis 3.2 hold. This hypothesis is not an additional assumption beyond Hypothesis 3.2; it is included only to make the appendix proof dependencies explicit.

**Theorem D.2** (Formal conditional hardness theorem). *Assume Hypothesis 3.2 (equivalently, its appendix restatement Hypothesis D.1). Then Construction 3.3 is a randomized polynomial-time samp!able map from the underlying average-case sparse-recovery input to multi-environment SC samples. Moreover, the resulting SC family has:*

1. *a unique invariant-predictive maximizer of the score $S(V)$;*
2. *polynomial-sample recovery by exhaustive search;*
3. *no polynomial-time constant-accuracy recovery algorithm unless Hypothesis 3.2 fails.*

**Theorem D.3** (Formal minimax rate theorem). *Under the locally regular Gaussian SC class described in Assumption 4.10, with diversity bounded below by $\gamma_0 > 0$, the minimax squared subspace risk satisfies*

$$\inf_{\hat{V}} \sup_{\mathbb{P}} \mathbb{E}_{\mathbb{P}}\left[\text{dist}(\hat{V}, V_{\text{inv}})^2\right] = \Theta\left(\frac{k(d - k)}{n|\mathcal{E}|}\right),$$

*with constants depending on the Gaussian regularity, covariance condition number, SNR, and local curvature constants.*

**Theorem D.4** (Formal diversity-dependent transition). *In the label-induced shift class satisfying Assumption 4.14, the*

*refined upper rate is*

$$\mathbb{E}\left[\text{dist}(\hat{V}, V_{\text{inv}})^2\right] \lesssim \frac{k(d-k)}{n|\mathcal{E}|\gamma^2}.$$

*This is the refined phase-transition rate stated in the main text. In the constant-diversity regime $\gamma = \Theta(1)$, it reduces to the usual parametric rate $k(d-k)/(n|\mathcal{E}|)$.*

# E. Logical Status of the Supervised Sparse Primitive

## E.1. A covariance-estimator sanity check

**Lemma E.1** (Empirical covariance recovery in easy scalar regimes). *Let $(Z_i, Y_i)_{i=1}^N$ be i.i.d. centered jointly Gaussian samples with*

$$\text{Cov}(Z, Y) = \lambda v, \qquad \text{Cov}(Z) = \sigma_Z^2 I_{d_Z}, \qquad \text{Var}(Y) = \sigma_Y^2,$$

*where $v \in \mathbb{R}^{d_Z}$ is a unit vector. Define*

$$\hat{c} := \frac{1}{N} \sum_{i=1}^N Z_i Y_i.$$

*Then with probability at least $1 - \delta$,*

$$\|\hat{c} - \lambda v\|_2 \le C \sigma_Z \sigma_Y \sqrt{\frac{d_Z + \log(1/\delta)}{N}}.$$

*If $v$ is $s$-sparse with minimum nonzero coordinate at least $c_0/\sqrt{s}$, coordinate thresholding recovers the support when*

$$N \gtrsim \frac{\sigma_Z^2 \sigma_Y^2}{\lambda^2} s \log(d_Z/\delta).$$

*Proof.* Each coordinate $Z_{ij} Y_i - \mathbb{E}[Z_{ij} Y_i]$ is sub-exponential with parameter controlled by $\sigma_Z \sigma_Y$. Vector Bernstein gives the $\ell_2$ bound, and coordinatewise Bernstein plus a union bound gives the $\ell_\infty$ bound needed for thresholding. $\square$

*Remark* E.2 (Why the primitive is stated abstractly). Lemma E.1 shows that a bare scalar Gaussian model with exposed cross-covariance $\text{Cov}(Z, Y) = \lambda v$ can be easy in the corresponding sample regime. Therefore Hypothesis D.1 should not be interpreted as this exposed covariance model. The hardness reduction uses a black-box sparse predictive recovery primitive whose computational hardness is assumed directly.

# F. Complete Proofs

## F.1. Proof of Proposition F.1 (Proxy Approximation)

**Proposition F.1** (Proxy Approximation Conditions). *Consider a linear Gaussian model with standardized features. Suppose:*

 (i) *invariant coordinates have cross-environment correlation range at most $\delta_{\text{inv}}$;*
 (ii) *at least half of the spurious coordinates have population correlation range at least $\gamma_0$;*
(iii) *$n_{\min} \ge C \log d$.*

*Then with probability at least $1 - d^{-1}$,*

$$|\hat{\gamma} - \gamma_{\text{pop}}| \le C \sqrt{\frac{\log(d|\mathcal{E}|)}{n_{\min}}} + \delta_{\text{inv}},$$

*where*

$$\gamma_{\text{pop}} := \text{median}_j \left( \max_e \rho_j^{(e)} - \min_e \rho_j^{(e)} \right).$$

*Proof.* For each $j, e$, empirical correlations concentrate:

$$|\hat{\rho}_j^{(e)} - \rho_j^{(e)}| \le C \sqrt{\frac{\log(d|\mathcal{E}|)}{n_{\min}}}$$

uniformly over all $j, e$ with probability at least $1 - d^{-1}$. Therefore the empirical range

$$\hat{r}_j = \max_e \hat{\rho}_j^{(e)} - \min_e \hat{\rho}_j^{(e)}$$

satisfies

$$|\hat{r}_j - r_j| \leq 2C\sqrt{\frac{\log(d|\mathcal{E}|)}{n_{\min}}}.$$

The median is 1-Lipschitz under coordinatewise $\ell_\infty$ perturbations, yielding the result. The $\delta_{\mathrm{inv}}$ term accounts for residual invariant variation. $\square$

### F.2. Proof of Theorem 4.3

*Proof.* Define

$$\mathcal{Z} = \{V \in \mathrm{Gr}(k, d) : \mathcal{I}_{\mathrm{gap}}(V) = 0\}.$$

By the invariance condition, $V_{\mathrm{inv}} \in \mathcal{Z}$. If another $V' \neq V_{\mathrm{inv}}$ also lies in $\mathcal{Z}$, the invariance functional cannot distinguish the two. Conversely, if $\mathcal{Z} = \{V_{\mathrm{inv}}\}$, then for every $\epsilon > 0$, compactness of

$$\{V : \mathrm{dist}(V, V_{\mathrm{inv}}) \geq \epsilon\}$$

and continuity of Gaussian conditional KL imply a strictly positive population gap away from $V_{\mathrm{inv}}$. Hence $V_{\mathrm{inv}}$ is identifiable. $\square$

### F.3. Proof of Proposition 4.4 and Corollary 4.7

**Gaussian conditional parameters.** Let

$$A = X_{\mathrm{inv}}, \qquad B^{(e)} = X_{\mathrm{sp}}^{(e)}, \qquad p = d - k.$$

In the label-induced model,

$$Y = w^{*\top}A + \epsilon, \qquad B^{(e)} = \mu_{\mathrm{sp}}^{(e)}Y + \eta^{(e)}.$$

For a candidate subspace $V$ with orthonormal basis

$$Q = \begin{pmatrix} Q_A \\ Q_B \end{pmatrix},$$

define

$$T_e = Q^\top X^{(e)}, \qquad r_e = Q_B^\top \mu_{\mathrm{sp}}^{(e)}.$$

Then

$$T_e = (Q_A^\top + r_e w^{*\top})A + r_e \epsilon + Q_B^\top \eta^{(e)}.$$

The conditional law $Y \mid T_e$ is Gaussian with parameters

$$\beta_e = \Sigma_{T,e}^{-1} c_e, \qquad \sigma_{e|T}^2 = \sigma_Y^2 - c_e^\top \Sigma_{T,e}^{-1} c_e,$$

where

$$c_e = Q_A^\top w^* + \sigma_Y^2 r_e$$

and

$$\Sigma_{T,e} = I_k + Q_A^\top w^* r_e^\top + r_e w^{*\top} Q_A + \sigma_Y^2 r_e r_e^\top.$$

**Lemma F.2** (Generic non-cancellation). *For fixed $Q_B \neq 0$, the set of parameters for which $r_e \neq r_{e'}$ but*

$$\mathbb{P}_e(Y \mid Q^\top X) = \mathbb{P}_{e'}(Y \mid Q^\top X)$$

*is contained in a proper algebraic variety and therefore has Lebesgue measure zero.*

*Proof.* The conditional parameters are rational functions of $r_e, Q_A, Q_B, w^*, \sigma_\epsilon^2$. Equality of two conditional Gaussian laws gives rational equations; after multiplying by nonzero determinant factors, these become polynomial equations. These polynomials are not identically zero because one can choose $r_e = 0$ and $r_{e'} \neq 0$ so that $c_e \neq c_{e'}$ generically. Hence the degeneracy set is a proper algebraic variety. □

*Proof of Proposition 4.4.* Assume the generic non-cancellation condition of Lemma F.2. If $Q_B = 0$, then $V \subseteq V_{\text{inv}}$, and since both have dimension $k$, $V = V_{\text{inv}}$. If $Q_B \neq 0$, full rank of $\mathcal{F}$ implies that the environment differences span $V_{\text{sp}}$, so there exist $e, e'$ such that
$$Q_B^\top (\mu_{\text{sp}}^{(e)} - \mu_{\text{sp}}^{(e')}) \neq 0.$$
Thus $r_e \neq r_{e'}$. By generic non-cancellation, the conditional laws $Y \mid Q^\top X$ differ across environments, so $\mathcal{I}_{\text{gap}}(V) > 0$. Therefore $V_{\text{inv}}$ is the unique zero-gap subspace. □

*Proof of Corollary 4.7.* If all $\mu_{\text{sp}}^{(e)}$ are identical, then the joint law of $(X, Y)$ is identical across environments. Therefore $\mathcal{I}_{\text{gap}}(V) = 0$ for every $V$, so $V_{\text{inv}}$ is not identifiable. □

### F.4. Proof of Theorem 4.11 and Corollary 4.12

**Local chart.**  Near $V_{\text{inv}}$, write
$$Q_\Theta = \begin{pmatrix} I_k \\ \Theta \end{pmatrix} (I_k + \Theta^\top \Theta)^{-1/2}, \qquad \Theta \in \mathbb{R}^{(d-k) \times k}.$$

For small $\Theta, \Theta'$,
$$\text{dist}(V_\Theta, V_{\Theta'}) \asymp \|\Theta - \Theta'\|_F.$$
The local dimension is $q = k(d - k)$.

**Lower bound.**  Take a local packing of size $\log M \gtrsim q$ with pairwise distance $\epsilon$. Gaussian KL regularity gives
$$\text{KL}(P_{\Theta_i}^{\otimes N} \| P_{\Theta_j}^{\otimes N}) \leq CN\epsilon^2.$$

Choosing $\epsilon^2 \asymp q/N$ and applying Fano yields
$$\inf_{\hat{V}} \sup_{\mathbb{P}} \mathbb{E}[\text{dist}(\hat{V}, V_{\text{inv}})^2] \gtrsim \frac{q}{N}.$$

**Upper bound via LAN.**  Under local Gaussian LAN regularity, the local MLE satisfies
$$\hat{\Theta} - \Theta_0 = I(\Theta_0)^{-1} \frac{1}{N} \nabla_\Theta L_N(\Theta_0) + o_{\mathbb{P}}(N^{-1/2}),$$
where
$$cI_q \preceq I(\Theta_0) \preceq CI_q.$$
Therefore
$$\mathbb{E}\|\hat{\Theta} - \Theta_0\|_F^2 \leq C\frac{q}{N}.$$
Using the local chart equivalence gives
$$\mathbb{E}[\text{dist}(\hat{V}, V_{\text{inv}})^2] \leq C\frac{k(d-k)}{n|\mathcal{E}|}.$$
The entropy term $\mathcal{C}_{\text{sp}}$ enters when model selection among approximately invariant local basins is required.

*Proof of Corollary 4.12.*  Set the upper bound to be at most $\epsilon^2$, yielding
$$n \gtrsim \frac{k(d-k)}{|\mathcal{E}|\epsilon^2}$$

up to logarithmic and regularity-dependent factors. □

## F.5. Proof of Theorem 4.15

**Proposition F.3** (Refined phase transition). *Under Assumptions 2.2, 4.10, and 4.14,*

$$N\gamma^2 \lesssim k(d-k) \implies \inf_{\hat{V}} \sup_{\mathbb{P}} \mathbb{E}[\text{dist}(\hat{V}, V_{\text{inv}})^2] \geq c,$$

*while*

$$N\gamma^2 \gtrsim k(d-k) \implies \exists \hat{V} : \mathbb{E}[\text{dist}(\hat{V}, V_{\text{inv}})^2] \leq C\frac{k(d-k)}{N\gamma^2}.$$

*Proof.* The label-induced model gives a local KL expansion

$$\text{KL}(P_\Theta \| P_{\Theta'}) \asymp \gamma^2 \|\Theta - \Theta'\|_F^2.$$

The lower bound follows from Fano using a constant-radius packing when $N\gamma^2 \lesssim k(d-k)$. The upper bound follows from the LAN argument with Fisher information lower bounded by $c\gamma^2 I_q$, yielding squared error $O(q/(N\gamma^2))$. $\quad\square$

*Proof of Theorem 4.15.* The critical sample size

$$n^* \asymp \frac{k(d-k)}{|\mathcal{E}|\gamma^2}$$

is equivalent to $N\gamma^2 \asymp k(d-k)$. The subcritical lower bound follows from Proposition F.3 when $N\gamma^2 \lesssim k(d-k)$. The supercritical upper bound follows from Proposition F.3 when $N\gamma^2 \gtrsim k(d-k)$, giving the refined rate

$$\mathbb{E}[\text{dist}(\hat{V}, V_{\text{inv}})^2] \leq C\frac{k(d-k)}{N\gamma^2} = C\frac{k(d-k)}{n|\mathcal{E}|\gamma^2}.$$

In the constant-diversity regime $\gamma = \Theta(1)$, this further simplifies to $O(k(d-k)/(n|\mathcal{E}|))$. $\quad\square$

## F.6. Samplable Sparse-Recovery Primitive and Embedding Details

This section clarifies how Hypothesis D.1 is embedded into the SC instance.

**Lemma F.4** (TV transfer). *If two dataset distributions are within $\varepsilon$ in total variation, then every algorithm's success probability differs by at most $\varepsilon$ under the two distributions.*

*Proof.* Total variation contracts under Markov kernels, and randomized algorithms are Markov kernels. $\quad\square$

**Samplability.** Given $A$, Construction 3.3 runs $\mathcal{R}(A)$ independently in each environment and then samples

$$W^{(e,t)} = \mu^{(e)} Y^{(e,t)} + \eta^{(e,t)}$$

using public randomness. The hidden sparse direction and its support are never used.

**SC validity.** Construction 3.3 satisfies the SC conditions as in Lemma 3.4: by Hypothesis 3.2, $Y \mid X_{\text{inv}}$ depends only on $v^\top Z$ and is identical across environments, while $W \mid X_{\text{inv}}$ varies with $\mu^{(e)}$. The sampler-generated construction inherits the recovery and hardness statements up to the total-variation discrepancies specified by the primitive, when applicable.

## F.7. Proof of Lemma 3.5

Let $u = (u_Z, a)$ be a unit vector spanning $V$, where $u_Z \in \mathbb{R}^{d_Z}$ and $a \in \mathbb{R}$. Let $u^* = (v, 0)$ span $V_{\text{inv}}$, and write $r = \|u_Z\| = (1 - a^2)^{1/2}$. If $r > 0$, define $\bar{u}_Z = u_Z/r$.

For the true direction,

$$S(V_{\text{inv}}) = A(V_{\text{inv}}) = \Phi_P(v), \qquad T(V_{\text{inv}}) = 0.$$

Since $0 \leq \Phi_P \leq 1$ and the primitive margin is positive, $\Phi_P(v)$ is bounded below by a positive constant depending on $c_{\text{prim}}$.

We first compute the invariance penalty. In Construction 3.3,

$$u^\top X = u_Z^\top Z + aW^{(e)} = u_Z^\top Z + a\mu^{(e)}Y + a\eta^{(e)}.$$

The primitive samples $(Z, Y)$ have the same distribution in every environment, and $\eta^{(e)}$ is independent of $Y$. Hence

$$\theta_V^{(e)} = \mathrm{Cov}_e(Y, u^\top X) = \mathrm{Cov}(Y, u_Z^\top Z) + a\mu^{(e)}\mathrm{Var}(Y).$$

Therefore, for any pair $e, e'$,

$$\theta_V^{(e)} - \theta_V^{(e')} = a(\mu^{(e)} - \mu^{(e')})\sigma_Y^2,$$

and consequently

$$T(V) = \max_{e,e'}(\theta_V^{(e)} - \theta_V^{(e')})^2 = a^2\Delta_\mu^2\sigma_Y^4.$$

Fix the target separation radius $\delta \in (0, 1)$. Choose a small constant $\alpha = \alpha(\delta) > 0$. We split into two cases.

**Case 1: $|a| \geq \alpha$.** Since $A(V) \leq 1$,

$$S(V) \leq 1 - \lambda\alpha^2\Delta_\mu^2\sigma_Y^4.$$

Choosing $\lambda$ sufficiently large, depending only on $\alpha, \Delta_\mu, \sigma_Y^2$, ensures

$$S(V) \leq S(V_{\mathrm{inv}}) - c_1$$

for some constant $c_1 > 0$.

**Case 2: $|a| < \alpha$.** For $\alpha$ sufficiently small as a function of $\delta$, the condition $\mathrm{dist}(V, V_{\mathrm{inv}}) \geq \delta$ implies that $\mathrm{span}(\bar{u}_Z)$ is bounded away from $\mathrm{span}(v)$ by a constant $\delta' = \delta'(\delta) > 0$. By the primitive margin,

$$\Phi_P(\bar{u}_Z) \leq \Phi_P(v) - c'_{\mathrm{prim}}$$

for some $c'_{\mathrm{prim}} > 0$ depending only on $c_{\mathrm{prim}}$ and $\delta'$. Since $r^2 \leq 1$ and $T(V) \geq 0$,

$$S(V) = r^2\Phi_P(\bar{u}_Z) - \lambda T(V) \leq \Phi_P(v) - c'_{\mathrm{prim}} = S(V_{\mathrm{inv}}) - c'_{\mathrm{prim}}.$$

If $r = 0$, then $A(V) = 0$ by definition and the same conclusion is immediate.

Combining the two cases gives

$$S(V_{\mathrm{inv}}) - \sup_{\mathrm{dist}(V, V_{\mathrm{inv}}) \geq \delta} S(V) \geq c_{\mathrm{mar}},$$

where $c_{\mathrm{mar}} = \min\{c_1, c'_{\mathrm{prim}}\} > 0$. This proves the claim.

### F.8. Proof of Lemma 3.6

Take an $\epsilon$-net of $\mathrm{Gr}(1, d)$ of size at most $(C/\epsilon)^{d-1}$. The primitive concentration assumption gives uniform convergence of the empirical predictive score, and the bounded-moment condition gives uniform concentration of the wrapper covariance statistics over the net. With

$$n_{\min} \gtrsim \frac{d\log(C/\epsilon) + \log(|\mathcal{E}|/\delta)}{c_{\mathrm{mar}}^2},$$

up to primitive-dependent polynomial factors, the empirical score is uniformly within $c_{\mathrm{mar}}/4$ of the population score. The margin in Lemma 3.5 then implies that the empirical maximizer lies within constant distance of $V_{\mathrm{inv}}$.

### F.9. Proof of Lemma 3.7

For one-dimensional subspaces,

$$\|P_{\hat{V}} - P_{V_{\mathrm{inv}}}\|_F^2 = 2\sin^2\phi,$$

where $\phi$ is the principal angle. Hence

$$|\langle \hat{u}, (v, 0)\rangle|^2 \geq 1 - \delta_0^2/2.$$

Restricting $\hat{u}$ to the $Z$-block gives constant overlap with $v$, sufficient for the primitive decoder in Hypothesis D.1. Therefore any polynomial-time SC recovery algorithm would solve the underlying average-case sparse-recovery problem, contradicting the hardness clause in Hypothesis 3.2.

### F.10. Proof of Theorem 5.1

Theorem 5.1 is a summary of sufficient tractability templates. Representative formal versions are given below.

**Proposition F.5** (Sparse enumeration). *If the invariant subspace is axis-aligned and supported on a $k$-set with population score gap $\Delta$, exhaustive subset search recovers it with*

$$n|\mathcal{E}| \gtrsim \frac{k \log d + \log(1/\delta)}{\Delta^2}.$$

**Proposition F.6** (Spectral separation). *If an efficiently computable contrast matrix has eigengap $\Delta$ separating invariant and spurious eigenspaces, then Davis–Kahan gives*

$$\mathrm{dist}(\hat{V}, V_{\mathrm{inv}})^2 \lesssim \frac{\|\hat{M} - M\|_{\mathrm{op}}^2}{\Delta^2}.$$

**Proposition F.7** (Moment screening). *If invariant and spurious coordinates have population correlation-range separation $\Delta$, then empirical correlation screening succeeds when*

$$n_{\min} \gtrsim \frac{\log(d|\mathcal{E}|/\delta)}{\Delta^2}.$$

## G. Connection to PC-Based Sparse CCA Reductions

This section is motivational and does not serve as a formal derivation of Hypothesis D.1. The formal hardness assumption used in the paper is the black-box supervised sparse recovery primitive in Hypothesis 3.2.

Sparse CCA observes two Gaussian views $(Z, U)$ with

$$\mathrm{Cov}(Z, U) = \lambda v w^\top,$$

where $v, w$ are sparse. Standard Planted-Clique-based reductions establish hardness of sparse CCA in certain regimes.

A scalar-response supervised primitive may be heuristically related to sparse CCA by projecting $U$ to a scalar $Y = g^\top U$, which gives

$$\mathrm{Cov}(Z, Y) = \lambda(g^\top w)v.$$

However, this projection changes the signal strength and may enter regimes where simple covariance estimation succeeds. Therefore Hypothesis D.1 is used as a black-box conditional sparse predictive recovery assumption rather than derived directly from sparse CCA.

## H. Experimental Details

This section preserves the experimental details and labels referenced in the main text.

### H.1. Synthetic Data Generation

We generate synthetic data using the linear Gaussian construction described in the main text. The invariant features satisfy

$$X_{\mathrm{inv}} \sim \mathcal{N}(0, I_k), \qquad Y = \langle \mathbf{1}_k/\sqrt{k}, X_{\mathrm{inv}} \rangle + \epsilon.$$

Spurious features are generated with environment-dependent correlations with $Y$. The diversity parameter is controlled through the range of these correlations.

Algorithms include exhaustive subset search, invariance screening, greedy forward selection, beam search, and spectral baselines.

### H.2. Extended Results

This subsection provides additional empirical results supporting the main-text claims. Table 6 reports extra ColoredMNIST results across diversity levels, Table 7 summarizes worst-group accuracy as sample size increases, and Table 8 together with Fig. 3 shows the training dynamics of IRM on ColoredMNIST.

*Table 6.* ColoredMNIST: additional results across diversity levels and sample sizes.

| $\gamma$ | $N$ | Method | OOD Acc | Worst-Group |
|---|---|---|---|---|
| 0.0 | 2000 | ERM | $.820 \pm .012$ | $.540 \pm .025$ |
| 0.0 | 2000 | IRM | $.835 \pm .010$ | $.610 \pm .020$ |
| 0.4 | 1000 | ERM | $.885 \pm .012$ | $.690 \pm .020$ |
| 0.4 | 1000 | IRM | $.900 \pm .010$ | $.760 \pm .018$ |
| 0.8 | 2000 | ERM | $.920 \pm .008$ | $.760 \pm .018$ |
| 0.8 | 2000 | IRM | $.945 \pm .006$ | $.840 \pm .012$ |

Table 6 complements the main ColoredMNIST results by varying both diversity and sample size. When $\gamma = 0$, worst-group accuracy remains low, indicating that additional samples alone cannot fully resolve the lack of informative environment variation. As $\gamma$ increases, both ERM and IRM improve, with IRM benefiting more in worst-group accuracy, consistent with the role of environment diversity in identifying invariant structure.

*Table 7.* Worst-group accuracy versus sample size on real benchmarks.

| Dataset | Method | $N = 1k$ | $N = 5k$ | $N \geq 10k$ |
|---|---|---|---|---|
| CMNIST | ERM | $.60 \pm .03$ | $.76 \pm .02$ | — |
| CMNIST | IRM | $.62 \pm .03$ | $.84 \pm .01$ | — |
| Wbirds | ERM | $.62 \pm .04$ | $.74 \pm .03$ | $.79 \pm .02$ |
| Wbirds | IRM | $.68 \pm .03$ | $.79 \pm .02$ | $.83 \pm .02$ |
| Wbirds | GDRO | $.72 \pm .03$ | $.82 \pm .02$ | $.85 \pm .02$ |

Table 7 illustrates the sample-size effect on real benchmarks. Worst-group accuracy generally improves as $N$ increases, supporting the finite-sample transition picture in the main text. The gains are especially visible in the small-to-moderate sample regime, suggesting that invariant or group-aware methods are most brittle when both diversity and sample size are limited.

*Table 8.* ColoredMNIST IRM training dynamics. Here $\hat{\gamma}_{\mathrm{repr}}$ is computed on evolving IRM features and can decrease as color information is suppressed.

| Epoch | OOD Acc | Worst-Group | $\hat{\gamma}_{\mathrm{repr}}$ | Train Acc |
|---|---|---|---|---|
| 5 | .890 | .710 | .040 | .940 |
| 10 | .925 | .790 | .022 | .970 |
| 15 | .940 | .825 | .015 | .985 |
| 20 | .945 | .835 | .012 | .990 |
| 29 | .948 | .840 | .011 | .993 |

Table 8 shows that OOD and worst-group accuracy improve throughout IRM training, while $\hat{\gamma}_{\mathrm{repr}}$ computed on the evolving IRM representation decreases. This is not in conflict with the diversity diagnostic in the main text: there, $\hat{\gamma}_{\mathrm{repr}}$ is used on preliminary ERM features to assess whether environments expose spurious variation, whereas during IRM training a decreasing proxy can indicate that color-related spurious information is being suppressed in the learned representation.

Fig. 3 visualizes the same training behavior. OOD accuracy rises quickly, while worst-group accuracy improves more slowly and saturates later. This gap between average OOD performance and worst-group robustness is consistent with the main-text observation that invariant structure may be statistically useful before it is fully recovered by practical training objectives.

## I. Practical Application Guide

Fig. 4 summarizes the practical workflow suggested by our theory and experiments. The key diagnostic is to first estimate representation-level diversity; if diversity is low, collecting more diverse environments is prioritized, whereas if diversity is adequate but the sample size is below the predicted threshold, collecting more samples or reducing effective dimension is recommended.

The flowchart in Fig. 4 is intended as a heuristic decision aid rather than a formal algorithm. It operationalizes the two main quantities emphasized in the paper: diversity, estimated by $\hat{\gamma}_{\mathrm{repr}}$, and effective sample complexity, estimated through the

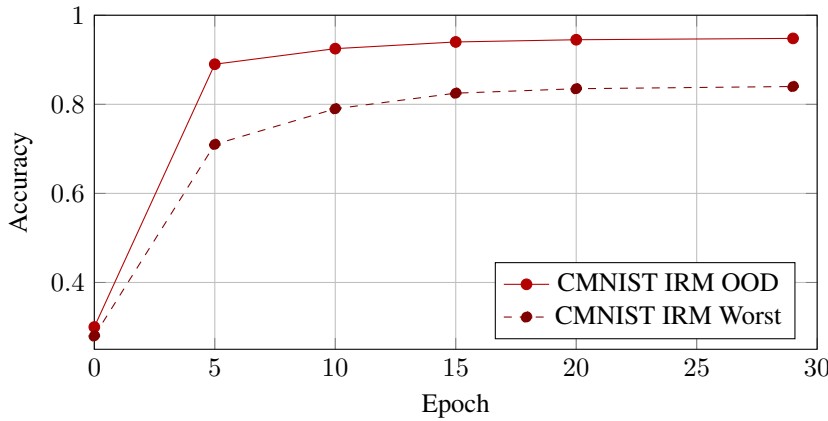

*Figure 3.* Training dynamics: worst-group accuracy lags behind OOD accuracy and saturates later.

scaling $n^* \propto k_{\text{eff}}(d_{\text{eff}} - k_{\text{eff}})/(|\mathcal{E}|\hat{\gamma}_{\text{repr}}^2)$. In practice, these estimates should be combined with validation performance and worst-group metrics when available.

**Step-by-step guide.**

1. Train a preliminary ERM model and compute $\hat{\gamma}_{\text{repr}}$ on penultimate features.

2. If $\hat{\gamma}_{\text{repr}}$ is small, prioritize collecting more diverse environments.

3. Estimate the critical scaling
$$n^* \propto \frac{k_{\text{eff}}(d_{\text{eff}} - k_{\text{eff}})}{|\mathcal{E}|\hat{\gamma}_{\text{repr}}^2}.$$

4. If $n \ll n^*$, collect more samples or reduce effective dimension.

5. Choose methods based on structure: GroupDRO when groups are available, IRM/REx when environments are informative, and spectral or screening methods when tractability conditions hold.

**Limitations.** The statistical theory is linear-Gaussian, and the computational component is worst-case and conditional on the black-box supervised sparse recovery primitive. The hardness result should be interpreted as a conditional lower bound for constructed hard families, not as a claim about typical benchmark datasets.

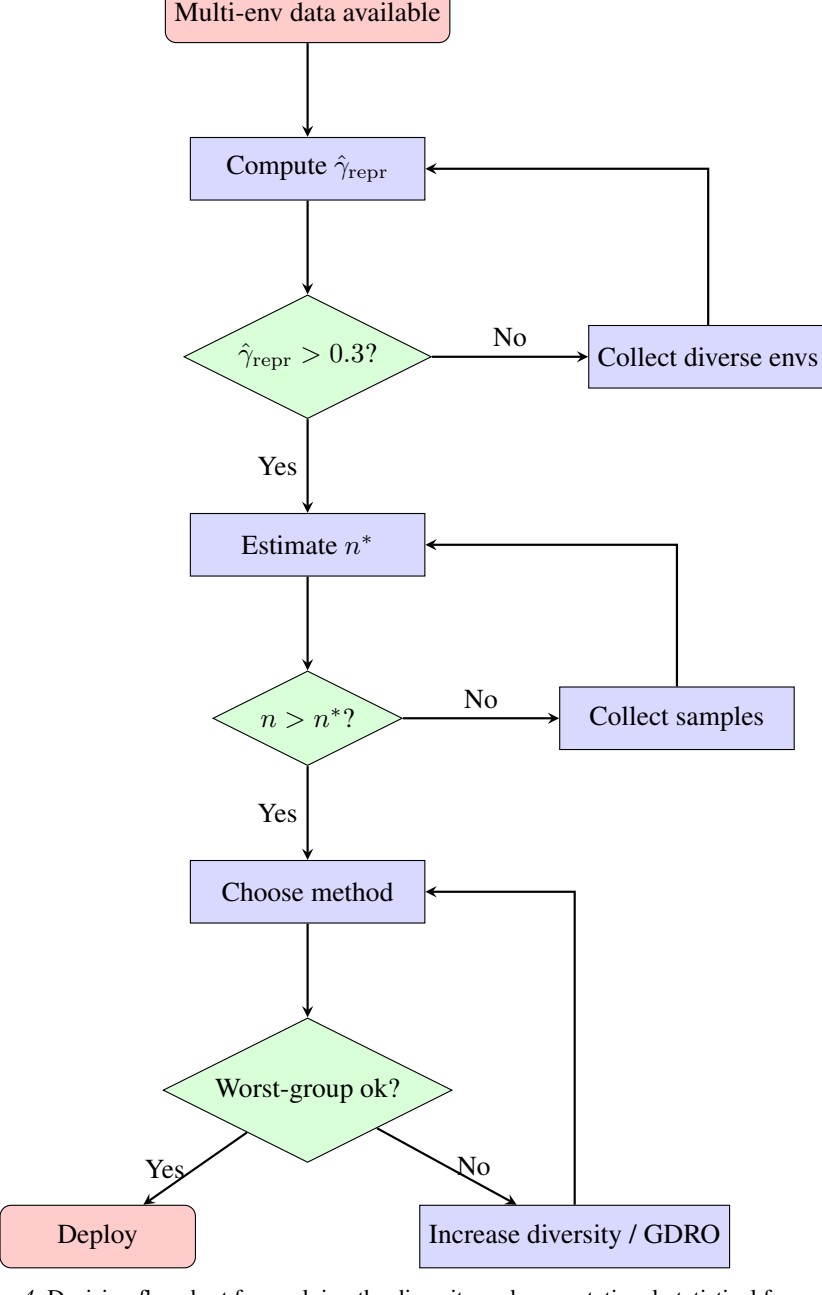

*Figure 4.* Decision flowchart for applying the diversity and computational-statistical framework.

