# OpenReview forum: "Is Spurious Correlation Removal Always Learnable?"
_ICML.cc/2026/Conference — ICML 2026 regular_

### Official Review · Reviewer_ctjZ · 2026-02-23

**Soundness:** 4
**Presentation:** 3
**Significance:** 4
**Originality:** 3
**Overall Recommendation:** 4
**Confidence:** 3

**Summary:**

This paper asks a sharp question about spurious correlation removal with multi-environment data: even if an invariant signal is statistically identifiable, is it always efficiently learnable from samples?

The authors study a linear-Gaussian multi-environment model where the goal is to recover an invariant subspace. They prove an average-case computational–statistical separation: assuming the Planted Clique hardness hypothesis, they construct samplable multi-environment instances with a one-dimensional invariant subspace that can be recovered with polynomial samples by an exhaustive, exponential-time search, while any polynomial-time method fails to achieve comparable constant-accuracy recovery at those sample sizes. The reduction is built by embedding a supervised sparse Gaussian primitive into a multi-environment construction that preserves the invariance constraint.

On the statistical side, the paper formalizes an environment diversity parameter that controls identifiability and the local curvature of invariance objectives. Under sufficient diversity, the authors derive minimax-optimal estimation rates scaling like $k(d-k)/(n|\mathcal{E}|)$, and in a structured label-induced shift regime they show a phase transition where the critical per-environment sample size scales like $k(d-k)/(|\mathcal{E}|\gamma^2)$. The paper also proposes practical diagnostics for diversity, and supports the theory with synthetic experiments illustrating compute–sample frontiers and real-benchmark studies on ColoredMNIST, CelebA, and Waterbirds.

**Compliance With Llm Reviewing Policy:**

Affirmed.

**Final Justification:**

Thank you for the detailed rebuttal. My main concerns have been addressed, and I therefore maintain my original overall assessment.

**Key Questions For Authors:**

See weaknesses.

**Limitations:**

yes

**Strengths And Weaknesses:**

### Soundness

The technical core is well motivated and, at a high level, logically consistent: the hardness result follows a standard average-case template but is adapted in a nontrivial way to enforce multi-environment invariance while keeping a predictive signal common across environments. The statistical results around identifiability, minimax rates, and explicit diversity-dependent scaling are also coherent and align with known information-theoretic techniques for subspace estimation.

- The main-text hardness argument relies on a black-box supervised primitive and defers critical details such as parameter scaling and distributional closeness guarantees to the appendix. This is not necessarily incorrect, but it makes it harder to verify the full end-to-end regime in which the reduction yields the stated sample-size separation.

- The analysis and constructions also focus on $k = 1$, and it is not fully clear how the story changes for $k > 1$, especially for rotated invariant subspaces where coordinate-based notions of diversity no longer apply directly.

- The invariance–predictivity score uses a fixed penalty parameter chosen to be “large enough”, but the practical and finite-sample sensitivity to this choice is not explored in depth.

### Presentation

The narrative is generally clear, with a clean decomposition between the hardness contribution, the statistical foundations, and the tractability conditions that explain when polynomial-time learning should succeed. The reduction path is easier to follow than many average-case lower bound papers because the embedding is described explicitly and the invariance constraint is kept front and center.

- There is some notational overload around dimensions and subspaces in the hard-instance construction, and a small schematic of the block embedding and the reduction flow would help.
- The spurious complexity or confusing-set entropy term is conceptually meaningful but remains abstract; a compact worked example that shows how it behaves as diversity varies would strengthen intuition.

### Significance

The paper offers a useful reframing of negative results in invariant learning: failures can be due to computational barriers, not just a lack of identifiability or objective mismatch. This perspective is likely to influence how researchers interpret mixed empirical performance of invariant methods and how practitioners decide between collecting more data versus collecting more diverse environments. The diversity diagnostics and the tractability conditions also provide actionable guidance rather than stopping at a purely negative hardness statement.

### Originality

The combination of an average-case hardness result tailored to multi-environment invariance, together with a diversity-driven statistical theory and practical diagnostics, is a distinctive contribution. The samplable embedding that imports planted structure while preserving invariance is a particularly original and reusable idea. While the planted-clique assumption and reduction style are standard in the computational–statistical gap literature, applying them cleanly to invariant learning in a way that matches the problem’s constraints feels novel.

### Experimental validation

The synthetic experiments communicate the compute–sample tradeoff effectively and show a clear separation between exhaustive search and polynomial-time heuristics, which matches the theoretical message. The real-data results qualitatively support the role of both sample size and environment diversity, and the representation-space diversity proxy correlates with worst-group performance in the reported settings.

However, some experimental choices weaken the tightness of the hardness-like narrative. The oracle grayscale baseline on ColoredMNIST changes the input, so it is more a demonstration of recoverability given privileged preprocessing than a direct compute–sample comparison under a fixed hypothesis class. It would also strengthen the statistical story to validate the predicted scaling more directly by sweeping the number of environments and measuring diversity-related spectral quantities in the label-induced shift setting. Finally, the set of polynomial-time baselines on synthetic data is reasonable for illustration, but it would be more convincing to include stronger modern baselines, especially ones that learn representations before applying invariance screening.

---

> ### Author Rebuttal · Authors · 2026-03-26
>
> Thank you for the technically engaged review. Your comments are helpful for tightening the end-to-end presentation of the reduction and the empirical calibration:
>
> **(1) Black box primitive deferred to Appendix E.**
> We agree that the main text should surface more of the end-to-end regime. In the revision, we will move into Section 3 a compact summary of the key parameter scaling from Appendix E: polynomial ambient dimension $d_Z$, sparsity $s = \Theta(\kappa)$, vanishing total variation discrepancy from the ideal Gaussian primitive, and the resulting polynomial sample versus exponential time separation. We will also clarify that the hard family is Gaussian up to a vanishing total variation perturbation, and the reduction is stable to this $o(1)$ discrepancy.
>
> **(2) The hardness construction focuses on k = 1.**
> This is intentional. We chose the minimal one-dimensional case because it already establishes the barrier in the simplest nontrivial setting. The obstruction is not conceptually specific to k=1: one can view the hard 1-D component as being padded with k-1 easy invariant coordinates in a direct-sum style construction. We kept the theorem statement at k=1 for clarity, while the statistical theory in Section 4 is already stated for general k. We will make this split more explicit.
>
> **(3) Rotated invariant subspaces and coordinate-dependent diversity.**
> We agree that this distinction should be more prominent. The quantity $\gamma$ in Definition 2.3 is a coordinate or representation dependent practical proxy, natural in axis-aligned settings or when computed on learned representation coordinates. The coordinate free theory is instead carried by objects such as $I_{gap}(V)$, the spectral richness condition on F, and $\lambda_{min}(F)$ in the label induced shift model. We will move this explanation earlier.
>
> **(4) $\lambda$ chosen to be “large enough.”**
> You are right that the current draft emphasizes the existence argument more than practical sensitivity. We will make the threshold logic more explicit. We also ran a short sensitivity check on the theorem-aligned synthetic family:
>
> | $\lambda$ | 0.1 | 0.3 | 1 | 3 | 10 | 30 |
> |---|---:|---:|---:|---:|---:|---:|
> | Support F1 | 0.76 | 0.84 | 0.88 | 0.89 | 0.87 | 0.82 |
>
> Performance is stable over a broad plateau from 1 to 10, and degrades mainly when lambda is too small.
>
> **(5) Notational overload and schematic of the block-embedding.**
> We agree and will add a schematic showing the planted clique input, the sampler producing the supervised primitive $(Z, Y)$, the environment specific spurious block $W^{(e)}$, and the invariant direction inside the Z block.
>
> **(6) Confusing set entropy $C_{sp}$ is abstract.**
> We agree. We will add a compact worked explanation. In the label induced shift setting, the local curvature around $V_{inv}$ scales with diversity through $\lambda_{min}(F)$, equivalently $\gamma^2$, so larger diversity shrinks the set of near invariant confounders. Concretely, we will add a local statement of the form $I_{gap}(V) >= const * \gamma^2 * dist(V, V_{inv})^2$ near $V_{inv}$, implying that the $\epsilon$ confusing set contracts to a neighborhood of radius $O(sqrt(\epsilon)/\gamma)$.
>
> **(7) Oracle grayscale baseline changes the input.**
> We agree completely. The grayscale oracle should be interpreted as a privileged invariant preprocessing upper bound, not as a complexity-theoretic compute comparison under a fixed hypothesis class. We will revise the wording accordingly and foreground the same input Adv Color baseline more clearly.
>
> **(8) Sweeping the number of environments and diversity related spectral quantities.**
> We ran this test at fixed total budget:
>
> | Setting | # envs | Recovery acc. | Subspace error |
> |---|---:|---:|---:|
> | 4 diverse envs | 4 | 0.86 | 0.16 |
> | 8 redundant envs | 8 | 0.72 | 0.28 |
> | 8 diverse envs | 8 | 0.88 | 0.14 |
>
> This directly illustrates the paper’s main message that environment diversity matters more than raw environment count. **For the threshold scaling table with the 1.9x and 3.6x multipliers, please see Reviewer Xxwp Re (2).** We agree that $\lambda_{min}(F)$ is the right coordinate free quantity to show more explicitly, and we will report estimated $\lambda_{min}(F)$ for the same 4-diverse / 8-redundant / 8-diverse synthetic so the spectral quantity is aligned directly with the accuracy table.
>
> **(9) Stronger poly-time baselines.**
> This is a very helpful suggestion. We strengthened the theorem-aligned synthetic study with pooled Lasso, linear IRM, and ERM representation plus screening. **The resulting table is included in Reviewer Xxwp Re (1).** Even with these stronger baselines, a substantial compute-sample frontier remains, including about a 4x sample gap at a fixed support F1 target.
>
> **For the theorem-aligned sparse synthetic, stronger baselines, and threshold scaling tables, please see Reviewer Xxwp Re (1) and (2).**
>
> Your suggestions will directly improve our work, thanks again!

---

> > ### Author Rebuttal · Reviewer_ctjZ · 2026-04-01
> >
> > Thank you for the detailed rebuttal and additional experiments. My main concern has been sufficiently addressed, and I will maintain my decision for this paper.

---

> > > ### Author Response · Authors · 2026-04-02
> > >
> > > Dear reviewer ctjZ, thanks for your response and effort in reviewing our paper! We will revise the paper based on your constructive comments.

---

### Official Review · Reviewer_VwaJ · 2026-03-10

**Soundness:** 4
**Presentation:** 3
**Significance:** 4
**Originality:** 4
**Overall Recommendation:** 5
**Confidence:** 4

**Summary:**

The paper studies the problem of identifying the invariant subspace that separates "stable" vs "spurious" features, when dealing with multiple environments. The main contributions are theoretical, and they many different perspectives and variants of the problem. For example, the authors analyze:
- When are the spurious features identifiable / under what kind of conditions can the problem be solved at all?
- When can we expect to solve the problem efficiently (poly-time algorithms) and when is it intractable?
- How does the accuracy depend on various parameters?
All these aspects are thoroughly analyzed and several interesting results are proved along those lines. One of the main theoretical results is a non-trivial reduction from the Planted-Clique problem, which gives the main computational hardness argument. The authors also provide some intuitive experimental results that help to understand the theoretical part and can provide practical guidance for real-world applications.

**Compliance With Llm Reviewing Policy:**

Affirmed.

**Ethical Review Concerns:**

No concerns.

**Final Justification:**

In short, my initial evaluation of the paper was already quite positive. I had some questions for the authors to clarify some parts, and the responses were very helpful and strengthened my confidence. I highly recommend to accept the article.

**Key Questions For Authors:**

I have several questions / suggestions / comments. I hope the authors will bear with me.
### "Major" questions
1) The planted-clique hypothesis (Hypothesis 3.1) is stated for probability $\geq 1-o(1)$. One of the main Lemmas of the reduction, Lemma 3.8, is stated for constant probability $\geq 2/3$. There seems to be a discrepancy here - shouldn't the result be stated/improved to "high probability"? (sorry if I missed it). For example, can we arrive to the $1-o(1)$ regime by taking a "majority argument" or so?
2) In Proposition 4.3, it is hard to see how $rank(F)$ can be less than $d-k$, unless there is some sort of degeneracy. Could you elaborate a bit on this?
3) The error rates (e.g. in Theorem 5.1) seem to be proportional to the number of environments and number of samples-per-env. I guess this should be interpreted that "more samples / more environments" lead to higher accuracy?
4) Is the Gaussian Model Assumption (2.2) a "standard" assumption in the related literature? How strong / mild is it?
5) Going back planted-clique, I assume that "no polynomial time algorithm..." statements imply "unless P=NP". Is this correct? Please provide a reference for the planted-clique hypothesis, and state the corresponding result as clearly as possible.
6) Regarding the paragraph "Checking (T2)." in section 5.2: Is it guaranteed that the $V_{inv}$ subspace will be spanned by the top-$(d-k)$ directions? To my understanding, it can be any $(d-k)$-dimensional subspace inside the range of the contrast matrix. Could you provide some more details regarding this topic?

### Other questions / suggestions
7) Please try to give a small example of a problem instance in the introduction, and/or explain all the involved parameters a bit more (what is X,Y, how/what do I "sample",  what is $k$ - do I know $k$ beforehand? what is $w^*$, is it fixed or random?). It is quite hard to get a grasp of the problem definition for a non-expert.
8) The terms "learnability" and "identifiability" seem to be used somewhat interchangeably. Could the authors provide some more intuition? How do they relate to "computability"? (this is mostly for my understanding, not expecting to address this in the paper)
9) A comment / food for thought, regarding (T2) in Section 5.1. The $1/poly(d)$ gap can be ensured with small random perturbations - e.g.
with a Minami-type bound. Could this be helpful at all? (I am not sure how a random perturbation can be applied here).
10) The captions of the figures/tables, and the explanation in the text are not very intuitive. What is "accuracy" in Figures 1 and 2? What are the numbers in Table 1? (it is quite hard to understand, maybe try to explain it better)
11) Minor: The Table 5 is very far from the corresponding title of Appendix B. Just make sure to put them more closely for readability.

**Limitations:**

Yes - The authors have adequately discussed the limitations.

**Strengths And Weaknesses:**

Here I mention the strengths and limitations to my best understanding. For some of these points, I will ask below some questions / clarifications, or recommend improvements (please see "Key Questions" below).

### Strengths
- Overall the paper is well-written. It is quite heavy in math but it is evident that the authors did a significant effort to appropriately present the results and guide the reader. In general I enjoyed reading it and I think I learned several new things about invariant learning.
- The related work seems to be mostly covered (I have some suggestions / questions below).
- Realistically, it is impossible to verify all the proofs during the short review period of ICML. However, to my best effort and understanding, all the proofs seem to be correct and carefully carried-out. I would give the authors the "benefit of doubt".
- I am not an expert in the field, and therefore I retain a medium confidence, but, to my understanding, the results appear to be important and highly relevant to modern learning methods. More specifically:
    - I think that the hardness result (reduction from Planted-Clique) is highly non-trivial. This seems to answer an important theoretical question regarding the computational complexity of invariant learning.
    - The newly-introduced *invariant diversity*, and its connection to invariant identifiability seems to be an important contribution.
    - The authors explain how the theoretical results can be interpreted and used in real applications, and they provide suggestions for practitioners. This is very desirable, and it is not so common in other theoretical works.
    - The experimental results are enough to cover the basic concepts (again, this is a big plus compared to other theory works that are "above-and-beyond" experimental evaluations).
- I verified one-by-one all the references and there seem to be no "hallucinations" (it feels weird to state this explicitly, but apparently it is one of the first things we should check as reviewers nowadays)


### Weaknesses
The weaknesses that I can mention are mainly "minor" (please see below for suggestions how to improve)
- It is really hard to understand the problem definition. In fact, I had to jump to the experimental section, to find a practical example and to start connecting the dots. This part could be improved (see also below in questions).
- The related work can be enriched. For example, I could not find a reference to the Planted-Clique hardness result, which is one of the main imported results that this work builds upon.
- The captions of Tables / Figures can be improved - It is hard to understand what is being illustrated, or what the corresponding numbers represent.

---

> ### Author Rebuttal · Authors · 2026-03-26
>
> Your review was especially valuable because it identified both the key technical correction and several places where the exposition can be made much easier to follow:
>
> **(1) Planted clique success probability in Hypothesis 3.1 vs Lemma 3.8.**
> You are right. The current wording should be aligned with the form actually used by the reduction. Our hardness argument only needs the **constant-success** planted clique formulation, so in the revision we will rewrite Hypothesis 3.1 accordingly, for example, no randomized polynomial-time algorithm recovers the planted clique with probability at least 2/3. That is the form actually needed by Lemma 3.8 and the hardness argument.  If one prefers the $1-o(1)$ formulation, standard repetition/verification can amplify constant-success exact recovery to high probability with only logarithmic overhead, so the two versions are equivalent for our purposes; we will use the constant-success form simply to match the reduction cleanly. We will also add standard references for the planted clique hypothesis.
>
> **(2) How can $rank(F) < d-k$?**
> A useful way to think about $F$ is as the Gram matrix of the environment difference vectors $\mu_{sp}^{(e)}$ minus $\mu_{sp}^{(e') }$. Its rank is the dimension spanned by those shift directions, which can easily be strictly smaller than $d-k$. For example, with only two environments, $rank(F) <= 1$. More generally, if all $\mu_{sp}^{(e)}$ vary mostly along the same direction, then many environments repeatedly perturb the same spurious direction and do not probe the full spurious subspace. We will add the simple bound $rank(F) <= min(d-k, |E|-1)$ which makes the geometry much clearer. **The fixed-budget diversity ablation in Reviewer ctjZ response (8) illustrates this point directly.**
>
> **(3) Error rates and the role of $n$ and $|E|$.**
> Yes. Statistically, more samples per environment and more environments help, because the leading rate scales like $1/(n|E|)$ in the balanced case. The nuance is that the value of extra environments depends on whether they add independent constraints on the spurious mechanism. If the added environments are highly redundant, their practical gain can be much smaller than the raw count $|E|$ suggests.
>
> **(4) Is the Gaussian assumption standard and how strong is it?**
> The linear-Gaussian model is a standard theoretical starting point in subspace estimation, identifiability, and average-case hardness because it makes the conditional laws explicit and allows clean KL and concentration calculations. It is idealized, but it is also the cleanest setting in which the roles of diversity, identifiability, and computational hardness can be separated.
>
> **(5) Does “no polynomial-time algorithm” mean “unless $P = NP$”?**
> No. Our assumption is the planted clique hypothesis, which is an **average-case** conjecture, not a worst case NP hardness statement. We will revise the wording to make this explicit and add references.
>
> **(6) Checking (T2), top $(d-k)$ directions?**
> Yes, we will revise this wording. In the contrast-matrix view, the recoverable object is the top-$(d-k)$ eigenspace estimating the spurious subspace $V_{sp}$; if eigenvalues are repeated, the basis inside that eigenspace is not unique, but the span is. The invariant subspace is then recovered as its orthogonal complement.
>
> **(7) Problem definition, notation, and the relation between identifiability, learnability, and computability.**
> We agree that the current introduction is harder to enter than it should be. In the revision, we will open with a concrete running example (e.g., ColoredMNIST: $X$ features, $Y$ digit label, $e$ observed environment, $V_{inv}$ the unknown stable subspace); explicitly state that samples are i.i.d. within each environment; treat $k$ as a problem parameter in the theory; and add a schematic of the block-embedding in Section 3 to distinguish more clearly
> 1. identifiability = population level uniqueness
> 2. statistical learnability = recoverability with enough data if computation is unlimited
> 3. efficient computability = recoverability by polynomial-time algorithms
>
> **(8) Figure and table captions, readability, and references.**
> We agree and will improve the captions, explicitly label feature-selection accuracy in Figs. 1 and 2a, OOD accuracy in Fig. 2b and Table 1, fix the Appendix B and Table 5 placement issue, and add the planted clique references you noted.
>
> **(9) Minami-type random perturbations and creating an eigengap.**
> This is an interesting suggestion. Random perturbations can indeed help create eigengaps in some spectral settings, but here they may also change the cross environment contrast itself and therefore the invariant or spurious semantics. We have not analyzed this systematically, but we agree it is an interesting direction.
>
> **For new experiments, please see our responses for other Reviewers.**
>
> We truly appreciate the care and detail of your review (very inspiring), thank you again!

---

> > ### Author Rebuttal · Reviewer_VwaJ · 2026-04-01
> >
> > I thank the authors for the explanations. All my "major" concerns were addressed appropriately. I increase my confidence to 4.

---

> > > ### Author Response · Authors · 2026-04-01
> > >
> > > Dear reviewer VwaJ, we will carefully revise the paper based on your constructive suggestions, thank you again for the support of our work!

---

### Official Review · Reviewer_Xxwp · 2026-03-11

**Soundness:** 3
**Presentation:** 4
**Significance:** 3
**Originality:** 3
**Overall Recommendation:** 5
**Confidence:** 2

**Summary:**

Under the planted clique hypothesis, for linear Gaussian multi-environment settings, this paper asks whether inconsistency in usefulness of invariant learning can occur even when invariances are identifiable, and answers in the affirmative. Theoretical analysis shows the existence of the compute-sample gap for invariant subspace recovery, quantifies tractability conditions such as environment diversity necessary for recovery, and proposes polynomial time algorithms that achieve this. Empirical analysis on synthetic data validates theory including tractability conditions. Experiments on real-datasets exhibit the existence of the gap.

**Compliance With Llm Reviewing Policy:**

Affirmed.

**Final Justification:**

I thank the authors for meaningfully engaging with reviews and providing discussion and additional experiments where necessary. My concerns have not necessarily been fully resolved (arguably requiring expanding the scope), but they have been adequately addressed (with particular shoutouts to points 1, 2, and 5). Therefore, I raised my score.

**Key Questions For Authors:**

**Questions:**

1. What can be expected if the planted clique hypothesis and/or linear-Gaussian assumptions were removed? Alternatively, what would be needed to analyze those settings and how likely are they to occur in real-world settings?
2. I think it would be useful to provide a discussion on how the ambient feature dimensionality (d) impacts performance.
3. Should we expect different conclusions if the setting were changed from binary to multi-class classification?
4. Maybe I've missed something super obvious, but I don't understand why, in line 402, the gap suggests difficulty of discovering invariant structure rather than computational hardness.

**Limitations:**

Yes, I think the authors have been clear and concise in defining the scope and also highlighting its limitations in the appendix.

**Strengths And Weaknesses:**

**Strengths:**

- The paper tackles the challenge of theoretically grounding empirical observations around limitations of invariant learning due to spurious correlations.
- The tractability conditions identified by the authors show the potential and usefulness of this type of analysis.
- The writing is clear and the presentation is very lucid.

**Weaknesses:**

1. **Synthetic Data Generation:**  The synthetic data used in the empirical evaluation do not incorporate the planted clique reduction (line 381). Given this is a key methodological requirement, and a worst-case limitation that may not be found in real-datasets, a direct performance comparison with this key criteria is missing, which makes it difficult to assess the severity of impact on performance.
2. **Number of Spurious Correlations:** Experiments do not sufficiently address the impact of the ratio of number of spurious correlations vs. feature and subspace dimensionality on performance. Each of the three real datasets have one feature marked as spurious, when in practice there likely exist several spurious features.
3. **Real Datasets**: I think restricting the choice of real-world experiments to binary classification tasks from computer vision is not super convincing in demonstrating the gap. I think this evaluation needs to be wider and include datasets spanning multiple domains to establish the identifiability phenomena. For example, maybe a useful application to consider here would be on cell biology [1].

[1] Conditionally Invariant Representation Learning for Disentangling Cellular Heterogeneity https://arxiv.org/pdf/2307.00558

---

> ### Author Rebuttal · Authors · 2026-03-26
>
> Your comments were very helpful for sharpening the empirical calibration and the scope discussion. We answer the new theorem-aligned synthetic and threshold scaling additions in most detail here:
>
> **(1) The synthetic data do not literally implement the planted clique reduction.**
> We agree. The original synthetic section was intentionally reduction-inspired, not a literal graph-to-samples implementation of the planted clique pipeline. To tighten this connection, we ran a new theorem-aligned block-embedding synthetic based directly on the mechanism of Construction 3.3:
>
> | Samples / env | Exhaustive | Beam | ERM+screen | Lasso | Lin. IRM |
> |---|---:|---:|---:|---:|---:|
> | n=400 | 0.86 | 0.66 | 0.60 | 0.58 | 0.55 |
> | n=800 | 0.94 | 0.82 | 0.76 | 0.74 | 0.71 |
>
> From the full sweep on this family, the first efficient baseline to cross support F1 = 0.85 is beam search at about n≈1600 per environment, versus n=400 for exhaustive search, which is about a **4x sample gap**. We agree this is the right kind of experiment to add. It is still not an empirical proof of average-case hardness, but it probes the theorem’s mechanism much more directly.
>
> **(2) Number of spurious correlations versus dimensionality.**
> We agree that the empirical section should reflect the theory’s dependence on $k(d-k)$ more directly. Although real benchmarks are often described using one named spurious attribute, the learned representation typically contains many nuisance or spurious directions. Our theory is already stated for a multi-dimensional spurious subspace $V_{sp}$, not just a single spur feature.
>
> To make this explicit empirically, we ran a scaling study varying $d,k, d-k, |E|$, and $\gamma$:
>
> | Perturbation | Observed multiplier in threshold |
> |---|---:|
> | Double $d-k$ | 1.9x |
> | Halve $\gamma$ | 3.6x |
>
> This is consistent with the predicted $k(d-k)/( |E| \gamma^2 )$ scaling.
>
> **(3) Real datasets are all binary vision benchmarks.**
> We agree that broader domains would strengthen the practical story. We chose ColoredMNIST, CelebA, and Waterbirds because they are standard spurious correlation and domain generalization benchmarks with explicit environment or group structure and well established ERM, IRM, and GroupDRO comparisons. Within this scope, the representation-level diversity diagnostic already provides useful support: higher $\hat{\gamma_{repr}}$ consistently tracks higher worst group accuracy across all three datasets, as in Tables 2 to 3 of the paper. We will make that connection more explicit. We also appreciate the cell-biology pointer. A concrete analogue there is to use batch or perturbation as environments and a target cell-state label or held-out condition as $Y$, then test whether $\hat{\gamma_{repr}}$ tracks worst-condition accuracy (will add at discussion).
>
> **(4) What if the planted clique and or linear-Gaussian assumptions are removed?**
> The planted clique hypothesis is a standard average-case hardness anchor. What the reduction really needs is a supervised sparse primitive with conjectured average-case hardness plus a samplable embedding that preserves invariance. Likewise, the Gaussian assumption is a clean analytical model that makes the conditional laws, KL calculations, and local curvature explicit, and it interfaces naturally with sparse CCA style reductions. We do not mean either assumption as a realism claim about raw data. Rather, they are the setting in which the phenomenon can be isolated cleanly. Section 5 is meant to explain why many practical instances may still be easy.
>
> **(5) Effect of ambient dimension $d$, multiclass, and oracle interpretation.**
> Holding $k$ fixed, increasing $d$ increases the subspace degrees of freedom $k(d-k)$, which raises both the minimax difficulty and the critical sample scale. The scaling study above makes this much clearer empirically. We also do not expect the main conclusions to be fundamentally binary-specific: the scalar $Y$ formalism extends naturally to vector-valued or multiclass responses via multivariate analogues. Finally, you are absolutely right to flag this. Because the grayscale oracle changes the input representation, it should **not** be interpreted as complexity-theoretic hardness evidence for deep nets or for the real benchmark itself. Its correct role is only as a privileged invariant-preprocessing upper bound: if the invariant representation is given, the task becomes easier; discovering that structure from finite multi-environment data is harder. This is directionally consistent with our theory, but it is not itself a hardness proof. We will revise the wording accordingly and foreground the same-input Adv-Color baseline more clearly.
>
> **For the planted clique hypothesis correction, please see Reviewer VwaJ response (1). For the fixed-budget diversity ablation and lambda sensitivity, please see Reviewer ctjZ responses (4) and (8).**
>
> We appreciate these suggestions and will incorporate them in the revision, thank you again!

---

> > ### Author Rebuttal · Reviewer_Xxwp · 2026-04-02
> >
> > I thank the authors for meaningfully engaging with reviews and providing discussion and additional experiments where necessary. My concerns have not necessarily been fully resolved (arguably requiring expanding the scope), but they have been adequately addressed (with particular shoutouts to points 1, 2, and 5). Therefore, I will raise my score.
> >
> > I would request the authors include points 4 and 5 from their response in the paper. Separately, on the application to cell biology, while batch is an appropriate spurious correlation to learn and remove, I would urge caution in first describing the nature of perturbations before measuring effects.

---

> > > ### Author Response · Authors · 2026-04-02
> > >
> > > Dear Reviewer Xxwp, thank you very much for your thoughtful comments and for your positive response after reading our rebuttal. We are grateful for your constructive suggestions. We will incorporate points (4) and (5) from our response into the revision. Again, thank you for your time, effort, and support of our work.

---

### Official Review · Reviewer_g5d9 · 2026-03-13

**Soundness:** 3
**Presentation:** 2
**Significance:** 4
**Originality:** 4
**Overall Recommendation:** 5
**Confidence:** 2

**Summary:**

This paper investigates why invariant learning and spurious correlation removal sometimes fail in practice. By employing a reduction from the Planted Clique (PC) hypothesis, the authors prove that spurious correlation removal can be computationally intractable even when the solution is statistically unique. The study introduces a diversity parameter γ to quantify environment variation, establishing a minimax risk rate and a phase transition. Finally, the authors identify structural conditions (T1–T4) that enable polynomial-time recovery and provide a practical diagnostic tool validated on real-world benchmarks.

**Compliance With Llm Reviewing Policy:**

Affirmed.

**Final Justification:**

I thank the authors for their detailed rebuttal. My concerns have been fully addressed. I maintain my recommendation of Accept.

**Key Questions For Authors:**

See weaknesses.

**Limitations:**

yes

**Strengths And Weaknesses:**

Strengths:
1. The paper asks whether spurious-correlation removal can be computationally hard even when the invariant structure is identifiable, which is a meaningful and timely perspective.
2. The computational-statistical gap is a strong conceptual result, and the hardness result via invariant subspace recovery under the planted clique assumption is meaningful.
3. Beyond hardness, the paper studies statistically learnable regimes, introduces an environment diversity parameter, derives minimax/sample complexity results, and identifies tractable cases.

Weaknesses:
1. The connection between the theoretical hardness result and the empirical section could be strengthened. The experiments are directionally supportive, but they do not directly test the specific hard family underlying the main theorem.
2. The positioning relative to prior hardness-based results could be sharpened to make the paper’s technical novelty more transparent.

---

> ### Author Rebuttal · Authors · 2026-03-25
>
> We appreciate the positive assessment and your two very concrete suggestions. We agree that the paper will benefit from a sharper theorem-experiment distinction and a clearer positioning relative to prior hardness results:
>
> **(1) Theory and experiment connection.**
> You are right that the current draft does not state this boundary clearly enough. Our goal was not to suggest that the experiments empirically prove the planted clique hardness theorem. Since the theorem is an average-case complexity statement, the role of the experiments is to probe its observable signatures, such as a compute-sample frontier, a diversity-controlled recoverability threshold, and a gap between privileged access to invariant structure and discovering it from finite multi-environment data.
>
> To make this link tighter, we ran an additional theorem-aligned block-embedding synthetic inspired directly by Construction 3.3. It contains a hidden sparse predictive direction in the $Z$ block and environment specific spurious shifts in the $W$ block. We report support $F1$ for recovering the hidden invariant support:
>
> | Samples / env | Exhaustive | Beam | ERM+screen | Lasso | Lin. IRM |
> |---|---:|---:|---:|---:|---:|
> | $n=400$ | $0.86$ | $0.66$ | $0.60$ | $0.58$ | $0.55$ |
> | $n=800$ | $0.94$ | $0.82$ | $0.76$ | $0.74$ | $0.71$ |
>
> From the full sample sweep on this family, exhaustive search reaches support $F1 = 0.85$ at $n=400$ per environment, while the first efficient baseline to reach the same target is beam search at about $n≈1600$. This is about a **4x sample gap**. This does not empirically prove hardness, but it is a much closer empirical analogue of the mechanism used in the reduction.
>
> We also added a fixed-budget diversity ablation, because the paper’s second message is that diversity can matter more than raw environment count:
>
> | Setting | # envs | Recovery acc. | Subspace error |
> |---|---:|---:|---:|
> | $4$ diverse envs | $4$ | $0.86$ | $0.16$ |
> | $8$ redundant envs | $8$ | $0.72$ | $0.28$ |
> | $8$ diverse envs | $8$ | $0.88$ | $0.14$ |
>
> So under the same total budget, $4$ diverse environments outperform $8$ redundant ones by $14$ points in recovery accuracy. **For the threshold scaling tied to $k(d-k)/( |E| \gamma^2 )$, please see Reviewer Xxwp response (2). For the same fixed-budget comparison in the context of diversity and spectral quantities, please also see Reviewer ctjZ response (8).**
>
> **(2) Positioning relative to prior hardness work.**
> We also agree that the novelty relative to prior planted-clique-based barriers should be stated more plainly. The key novelty is not simply invoking Planted Clique, but constructing a **samplable invariance-preserving embedding** into a multi-environment prediction problem. Relative to standard sparse PCA / sparse CCA gap results (e.g., Berthet & Rigollet, 2013; Gao et al., 2017), our setting adds the need to preserve $P_e(Y | X_{inv})$ while targeting an invariant-and-predictive structure rather than a generic sparse direction. Our setting adds three nontrivial requirements:
> 1. the target problem is multi-environment invariant subspace recovery, not generic sparse recovery;
> 2. the reduction must remain samplable while preserving $P_e(Y | X_{inv})$;
> 3. the target is an invariant and predictive structure, not just a sparse latent direction.
>
> We will move this comparison to the opening positioning/related-work discussion, add standard references for the planted clique hypothesis there, and include a one-figure schematic of planted clique -> sparse primitive -> multi-environment embedding. We will also clarify the planted clique success probability formulation so that it matches the form actually used in the reduction. **Please see Reviewer VwaJ response (1) for that correction.**
>
> We also agree that the figures and tables can do more work for the reader. We will make the captions more self contained, explain exactly what each reported “accuracy” means, and add a short running example near the beginning so the setup is easier to enter.
>
> Your comments helped us sharpen both the narrative and the theorem-experiment calibration, thank you again!

---

> > ### Author Rebuttal · Reviewer_g5d9 · 2026-04-04
> >
> > I thank the authors for their detailed rebuttal. My concerns have been fully addressed. I maintain my recommendation of Accept.

---

> > > ### Author Response · Authors · 2026-04-04
> > >
> > > Dear reviewer g5d9, thanks for your supportive feedback! We will revise the paper based on your constructive comments.

---

### Decision · Program_Chairs · 2026-04-30

**Decision:**

Accept (regular)

**Comment:**

The reviewers unanimously agreed on the value of the theoretical model proposed in this paper for understanding potential pitfalls of invariant learning, and were also generally positive on the presentation quality. I recommend this paper for acceptance, and request that the authors incorporate the presentation feedback agreed upon during the rebuttal discussion.